# Ice nucleating particle concentrations unaffected by urban air pollution in Beijing, China

Jie Chen[1], Zhijun Wu[1], Stefanie Augustin-Bauditz[2], Sarah Grawe[2], Markus Hartmann[2],

Xiangyu Pei[3], Zirui Liu[4], Dongsheng Ji[4], Heike Wex[2]

[1] State Key Joint Laboratory of Environmental Simulation and Pollution Control, College of Environmental Sciences and Engineering, Peking University, 100871, Beijing, China.

[2] Leibniz Institute for Tropospheric Research, 04318, Leipzig, Germany.

[3] Department of Chemistry and Molecular Biology, University of Gothenburg, 41296, Gothenburg, Sweden.

[4] State Key Laboratory of Atmospheric Boundary Layer Physics and Atmospheric Chemistry, Institute of Atmospheric Physics, Chinese Academy of Sciences, 100029, Beijing, China.

*Corresponding author:* Zhijun Wu (zhijunwu@pku.edu.cn)

**Key Points:**

       Ice nucleation

       Urban aerosol

       Immersion mode

**Abstract**

Exceedingly high levels of $PM_{2.5}$ with complex chemical composition occur frequently in China. It has been speculated if anthropogenic $PM_{2.5}$ may significantly contribute ice nucleating particles (INP). However, few studies have focused on the ice-nucleating properties of urban particles. In this work, two ice-nucleating droplet arrays have been used to determine the atmospheric number concentration of INP ($N_{INP}$) in the range from -6 ℃ to -25 ℃ in Beijing. No correlations between $N_{INP}$ and neither $PM_{2.5}$ nor black carbon mass concentrations were found, although both varied by more than a factor of 30 during the sampling period. Similarly, there were no correlations between $N_{INP}$ and either total particle number concentration or number concentrations for particles with diameters > 500 nm. Furthermore, there was no clear difference between day and night samples. All these indicate that Beijing air pollution did not increase or decrease INP concentrations in the examined temperature range above values observed in non-urban areas, hence, the background INP concentrations might not be anthropogenically influenced as far as urban air pollution is concerned, at least in the examined temperature range.

**1 Introduction**

Formation of the ice phase in clouds can be modulated by aerosols emitted from anthropogenic and natural sources (Morris et al., 2014; Murray et al., 2012; Rosenfeld et al., 2008) via heterogeneous ice nucleation (Pruppacher et al., 1998). This results in a significant impact on the cloud extent, lifetime, formation of precipitation, and radiative properties of clouds (DeMott et al., 2010). Currently, four mechanisms are proposed for the heterogeneous ice nucleation in mixed-phase clouds: deposition ice nucleation, condensation freezing, immersion freezing, and contact freezing (Vali et al., 2015; Hoose and Möhler, 2012). It is under discussion if condensation freezing is different from immersion freezing on a fundamental level (Wex et al., 2014) and if at least some of the observed deposition ice nucleation can be attributed to pore condensation and freezing (Marcolli, 2014). For mixed-phase clouds, immersion freezing has been widely reported to be the most important ice nucleation mechanism (Ansmann et al., 2008; Murray et al., 2012; Westbrook and Illingworth, 2013). During the past decades, great efforts have been dedicated to understanding heterogeneous ice nucleation. However, it has become obvious that many fundamental questions in this field are still unsolved (Kanji et al., 2017).

Numerous studies have attempted to quantify the ice nucleation ability of selected aerosol particles
of a specific composition in immersion mode, such as dust (DeMott et al., 2015; Kaufmann et al., 2016;
DeMott et al., 2003), marine (Wilson et al., 2015; DeMott et al., 2016; Alpert et al., 2011) and biological
particles (Pummer et al., 2012; Hartmann et al., 2013; Fröhlich-Nowoisky et al., 2015). Szyrmer and
Zawadzki (1997), Hoose and Möhler (2012), Murray et al. (2012) and Kanji et al. (2017) are all reviews
which give a more extensive overview over materials that can induce ice nucleation. In general, biogenic
particles have been assumed to provide atmospheric ice nucleating particles (INP) which are ice active
in the immersion mode at comparably high temperatures (above -15°C, Murray et al., 2012; Petters &
Wright, 2015). Ice activity at lower temperatures is attributed to mineral dust particles (Murray et al.,
2012) while the role of soot particles in atmospheric ice nucleation is still debated (Kanji et al., 2017).
Biogenic particles in general have long been known to be able to induce ice nucleation at
comparably high temperatures above -10°C (e.g. Schnell and Vali, 1972). It has been widely accepted
that biological particles can act as efficient INP, with some bacteria and fungi reported to possess the
ability to arouse freezing at temperatures as high as -2°C to -5°C (Lundheim, 2002). Fungal spores
(O'Sullivan et al., 2016; Pummer et al., 2015) and lichen (Moffett et al., 2015) are known to nucleate ice
in the temperature range above -10°C, while pollen (Augustin et al., 2013; Pummer et al., 2012) may
compete with mineral dust particles in terms of their ability to nucleate ice, albeit not in terms of their
atmospheric abundance.
Recognized as the dominant INPs in mixed-phase clouds (Kamphus et al., 2010), particles from
various mineral dusts were found to catalyse ice formation effectively in chamber experiments (Murray
et al., 2012; Kanji et al., 2017). Among mineral dust particles, those containing K-feldspar might be
particularly ice active (Atkinson et al., 2013).
In general, burning of liquid fuels produces soot particles (i.e., particles that are mostly organic),
while burning of solid material as e.g., biomass or coal will also produce ash particles which contain the
inorganic components that made up the fuel. Umo et al. (2015) and Grawe et al. (2016) examined the ice
activity of ash particles from wood and coal burning in the immersion mode and both find that these
particles are ice active. In Grawe et al. (2016), ash particles with atmospherically relevant sizes of 300 nm
were examined and the most active particles came from a sample of fly ash from a coal burning power
plant, inducing immersion freezing below -22°C. Both, Umo et al. (2015) and Grawe et al. (2016) suggest
that ash particles might play a role in the atmosphere, however, point to a lack of knowledge of their
atmospheric abundance. Also, different ash samples showed different ice activities, and also large
differences in the results between the methods used for the examination were described, i.e., it is still
inconclusive if ash particle might play an important role as atmospheric INP.
Although there has been a considerable number of studies aimed at understanding the ability of
black carbon (BC)-containing particles acting as INP, the results are still controversial. Some studies
show that BC-containing particles did not act as good INP (Schill et al., 2016; Chou et al., 2013). Chou
et al. (2013) observed that soot particles from diesel engines and wood burning form ice at -40°C, and
unrealistically high relative humidity (RH) was needed for freezing initiation above this temperature.
Schill et al. (2016) showed that neither fresh nor aged emissions from diesel engines contributed
appreciably to atmospheric INP concentrations. However, some studies considered BC-containing
particles as possible INPs (Cozic et al., 2008; Levin et al., 2016; Cozic et al., 2007). Observation of
abundant BC in ice particle residuals in field experiments suggested that some BC-containing particles
may preferentially act as INP (Cozic et al., 2008). In the experiments conducted by Levin et al. (2016),
emissions of different types of biomass fuel produced measurable concentrations of INPs (0.1-10 cm$^{-3}$)
associated with higher BC concentration accounting for about 0-70%. Determination of ice nucleating
properties of physically and chemically aged soot particles suggests that the heterogeneous ice nucleation
activity of freshly emitted diesel soot particles is sensitive to some of the aging processes (Kulkarni et
al., 2016).
In the atmosphere of urban areas with dense population, various sources and complex aging
transformations (such as coagulation, condensation of vapor, chemical reaction) of particles can be found.
Particularly, urban aerosol may be rich in BC-containing particles resulting from anthropogenic activities,
such as fossil fuel combustion and biomass burning (Bond et al., 2013), which were speculated to play a
role for the formation of ice in clouds (Kanji et al., 2017). However, the ice nucleating properties of
particles produced in urban regions have rarely been the focus of previous studies. Exceptions are Knopf
et al. (2010) and Corbin et al. (2012), examining the ice nucleation activity of particles in the
anthropogenically influenced atmospheric aerosol in Mexico City and Toronto, respectively. In both
studies the relative humidity at which measurements were made were below water vapor saturation (with
respect to liquid water). Using filter samples, Knopf et al. (2010) state that organic particles included in
their samples might potentially induce ice nucleation at conditions relevant to cirrus formation. Corbin
et al. (2012) used a CFDC (Continuous-Flow Diffusion Chamber) operating at -33°C together with a
particle mass spectrometer. Statistical limitations impeded a statistical sound analysis, but their data
suggests that dust particles, particles from biomass burning and particles containing elemental carbon
might be sources of INP at their experimental conditions. They explicitly encourage further studies of
these particles types concerning their role as possible INP.
In the present study, we measured the ice nucleating activity of urban aerosols in parallel with BC
and $PM_{2.5}$ mass concentration and particle number concentrations in the atmosphere of the mega-city
Beijing, which is frequently experiencing heavy pollution. During heavy haze episodes, $PM_{2.5}$ mass
concentration can be several hundred micrograms per cubic meter and composed of a complex mixture
of different chemical components (organic matter, inorganic ions and black carbon) (Zheng et al., 2016).
The goal of this project is to find out if anthropogenic sources which are dominant in the urban
atmosphere significantly contribute to the local INP concentrations, focusing particularly on the ice
nucleating ability of BC in urban aerosols.
**2 Materials and Methods**
**2.1 Sample collection and particle number measurement**
The sampling site for particle collection was on the roof of a six-floor building (about 30 m above
ground level) on the campus of Peking University (39°59′20″N, 116°18′26″E), located in the north-
western urban area of Beijing.
Particles with an aerodynamic diameter less than or equal to 2.5 micro-meters ($PM_{2.5}$) were collected
on quartz fiber (Whatman, 1851-865) and PTFE filters (Whatman,7592-104) using a 4-channel sampler
with 2.5μm impactors from 27th November 2016 to 1st December 2016 and 13th December 2016 to 22th
December 2016. Daytime filters were collected from 8:00 am to 8:00 pm and nighttime filters were
collected from 8:00 pm to 8:00 am with an air flow rate of 16.7 L min$^{-1}$, resulting in a total volume of air
sampled on each filter of ~12000 L. Note that all sample volumes used herein were converted to standard
volumes. The quartz filters were treated before the sampling by heating them to 550 ℃ for 6 h. After
sampling, all filters were kept at ≤ -18 ℃ during storage, and the INP analysis was done within 20 days,
starting on 5th February in 2017.
A scanning mobility particle sizer (SMPS, TSI Inc., USA) system was used to obtain particle
number distribution in the 3-700 nm (electrical mobility diameter) size range during the sampling period
while an aerodynamic particle sizer (APS, TSI model 3321, TSI Inc., USA) measured particle number
size distributions between 800 nm and 2.5μm (aerodynamic diameter). The APS results were transformed
from aerodynamic diameter to Stokes diameter with a particle density of 1.5 g cm$^{-3}$ which were measured
by a CPMA (centrifugal particle mass analyzer) and combined with the measured and inverted size
distributions obtained from the SMPS. From these combined size distributions, we calculated the total
particle number concentration of particles in the diameter range from 3nm to 2.5μm ($N_{total}$) and number
concentrations of particles larger than 500nm ($N_{>500nm}$). When comparing with filter results, we use 12h-
average values of $N_{total}$ and $N_{>500nm}$., where the averages were always made from 8:00 am to 8:00 pm for
daytime data and from 8:00 pm to 8:00 am for nighttime data. $N_{>500nm}$ was derived, as in general larger
particles are expected to be more efficient INP, and also as this size range was selected in DeMott et al.
(2010, 2015) to serve as a base for parameterizations of INP number concentrations.
Concentrations of BC were continuously measured by a multi-angle absorption photometer (5012
MAAP, Thermo Fisher Scientific, Waltham, MA, USA) utilizing a 637 nm LED as a light source (Müller
et al., 2011). The instrument measures the absorption of particles collected on a filter with a time
resolution of 5 min and automatically derives BC mass concentration from the measurement while
accounting for multiple scattering occurring on the filter. It might be worth noting that a comparison of
BC concentrations obtained from the MAAP with concentrations of EC determined by a filter-based
SUNSET EC/OC analyzer during a different field campaign showed, that both instruments measured the
same trends while the mean ratio of concentrations of BC to EC was about 1.35.
**2.2 Chemical analysis**
Two PTEF filters were always sampled in parallel, and while one was used for INP analysis, the
other was selected for the total mass and water-soluble ion analysis. $PM_{2.5}$ mass concentration was
obtained with an analytical balance by the gravimetric method (Mettler Toledo AG285) (Yang et al.,
2011). As for water-soluble inorganic compounds analysis, Guo et al. (2012) described the method for
seven major ions ($K^+$, $Mg^{2+}$, $Ca^{2+}$, $NH_4^+$, $NO_3^-$, $SO_4^{2-}$ and $Cl^-$) measured by ion-chromatograph (DIONEX,
ICS-2500/2000) based on the usage of PTEF filters. Post-sampling, all filters were stored in the
refrigerator at -18 °C before analysis.

## 2.3 INDA and LINA analysis

Two devices called INDA (Ice Nucleation Droplet Array) and LINA (Leipzig Ice Nucleation Array) have been set up at the Leibniz Institute for Tropospheric Research (TROPOS) in Germany following the design described in Conen et al. (2012) and in Budke & Koop (2015), respectively. INDA was used to investigate the immersion freezing properties of the quartz fibre filter samples while LINA was used to test the particles on PTFE filters.

INDA consists of a thermostat (JULABO FP40) with a 16 L cooling bath. 96 circles (1mm in diameter each) of each quartz filter were cut out by a punch and immersed separately in the tubes of a PCR (Polymerase chain reaction) tray which each contained 50 µl distilled water. While Conen et al. (2012) originally used separate Eppendorf Tubes®, the use of PCR trays for immersion freezing studies has been suggested before in Hill et al. (2016) and was adapted in the LINA set-up. The PCR trays were placed on a sample-holder and exposed to decreasing temperatures with a cooling rate of approximately 1 K min$^{-1}$ in the cooling bath down to -30 °C. Real time images of the PCR trays were recorded every 6 seconds by a CCD (Charge Coupled Device) camera. A flat light that was fixed at the bottom of the cooling bath helped to yield proper contrast between frozen and unfrozen droplets on the recorded pictures, so that frozen droplets could be identified according to the brightness change during the freezing process. A program recorded the current temperature of the cooling bath and related it to the real-time images from the CCD camera. The temperature in the PCR trays had been calibrated previously as described in section 1.1 of the appendix.

For the measurement of ice nucleating particles at lower temperature, LINA was built according to an optical freezing array named BINARY, which was described in detail by Budke & Koop (2015). PTFE filters collected during the same period as quartz fibre filters were used for LINA. Half of the PTFE filter of each day was immersed in 10 ml distilled water and shaken for 1 h to wash particles off. For each measurement, 90 droplets with the volume of 1 µl were pipetted from the resulting suspension onto a thin hydrophobic glass slide (diameter 40 mm, thickness 0.13-0.16 mm, obtained from Marienfeld-superior), with each droplet being contained in a separate compartment. These compartments were round holes with diameters of 3 mm, drilled into an aluminium plate with a diameter of 40 mm and a thickness of 14 mm. Both, hydrophobic glass slide and the aluminium plate with the compartments were surrounded by an aluminium ring with an inner diameter of 40 mm, which acted to keep glass slide and

aluminium plate in place. Slide, plate and ring were all arranged before the droplets were pipetted. A
second thin glass slide was put on top of the plate so that the compartments were all separated from each
other and that evaporation of the droplets was prevented. The droplets were cooled on a Peltier element
with a cooling rate of 1 K min$^{-1}$. There was a thin oil (squalene) film between the hydrophobic glass slide
and the Peltier element for optimal heat conductivity. The temperature on the glass slide had been
determined previous to the experiments as described in section 1.2 of the appendix, and the temperature
shift between that set on the Peltier element and that observed on the glass slide was accounted for in the
data presented herein. The freezing process again was recorded by taking pictures with a CCD camera
every 6 seconds and detecting the freezing based on a change in the reflectance of the droplets upon
freezing.
As mentioned above, the temperature calibration for these two instruments is described in detail in
the section 1.1 and 1.2 of the appendix. The background freezing signal of pure distilled water and circles
cut from clean filters were tested as well. These results are shown in the section 2 of the appendix.
The measurements resulted in frozen fractions ($f_{ice}$) as defined in Eq. (1):
$$f_{ice} = \frac{N_{frozen}}{N_t} \qquad (1)$$
where $N_{frozen}$ is the number of frozen tubes or droplets at a certain temperature and $N_t$ is the total number
of tubes in PCR trays (i.e., 96) or droplets on the slides (i.e., 90).
The temperature dependent cumulative number concentration of INP ($N_{INP}$) per volume of sampled
air was calculated according to Eq. (2), similarly to Vali (1971) and Conen et al. (2012):
$$N_{INP}(T) = -\frac{\ln(1 - f_{ice}(T))}{V_{sampled}} \qquad (2)$$
where $N_{unfrozen}$ is the number of tubes or droplets still unfrozen (liquid) at a certain temperature, and
$V_{sampled}$ is the volume of air converted to standard conditions (0°C and 1013hPa) from which the particles
were collected that were suspended in each of the droplets in LINA or that were collected on each filter
punch used for INDA measurements, respectively.
The chemical ion analysis in section 3.1 and the determination of the PM$_{2.5}$ mass concentration was
done at Peking University. The filters used for INP measurements were brought to TROPOS where then
INP measurements were done. Filters were continuously cooled below 0°C in a portable ice box during
transport.

**3 Results and Discussion**

**3.1 Severe PM$_{2.5}$ pollution in Beijing**

Fig. 1 shows the time series of PM$_{2.5}$ mass concentrations and chemical composition during the sampling period. The PM$_{2.5}$ mass concentration with a mean value of 97.30±77.9 μg m$^{-3}$ ranged from 6.54 μg m$^{-3}$ up to 273.06 μg m$^{-3}$. Here, the cases with PM$_{2.5}$ above 50 μg m$^{-3}$ were defined as polluted days, whereas the rest was defined as clean days. On average, the sulfate, nitrate, and ammonia (SNA) accounted for around 35% of PM$_{2.5}$ during the whole period with an obvious enhancement in polluted days (53%), indicating that generation of secondary particulate mass is one major contributor to the formation of particulate pollution, as it has previously been described in Guo et al. (2010) and Zheng et al. (2016). In this study, when we refer to secondarily formed particulate matter, this will always stand mainly for SNA and secondary organic substances. Dust particles are in the coarse mode, and only contribute little to the total PM$_{2.5}$ load (Lu et al., 2015; Li and Shao, 2009). In these studies, Ca$^{2+}$ as a tracer for dust particles showed a low proportion in PM$_{2.5}$, suggesting that the dust particles also only contributed little to PM$_{2.5}$ during our observations as well.

During the sampling period, BC mass concentrations varied from 0.50 μg m$^{-3}$ on clean days up to 17.26 μg m$^{-3}$ on polluted days. On average, the mean mass concentration of BC, 7.77±5.23 μg m$^{-3}$, accounted for about 13% of PM$_{2.5}$. During night time, BC concentrations were higher than those during daytime due to stronger diesel engine emissions and a lower boundary layer (Guo et al., 2012). Our previous studies showed that secondarily and primarily formed organic particulate matter contributed to around 36% of non-refractory PM$_1$ detected by an aerosols mass spectrometer during wintertime in the atmosphere of Beijing (Hu et al., 2017).

Additionally, Fig. 2 shows 2-day back-trajectories obtained by the NOAA HYSPLIT model, with one trajectory related to each sampled filter, starting at the median sampling time of each filter. Fig. 3 shows minutely recorded data for wind-direction and wind-speed collected by an Auto weather station (Met One Instruments Inc.) located on the same roof top as the aerosol sampling equipment. Both pictures are colored-coded with respect to PM$_{2.5}$ mass concentrations. The air masses that came from north or north-western directions were generally coincident with higher wind-speeds. They brought clean air with lower PM$_{2.5}$ mass concentrations. They did cross desert regions, however, Beijing was reported to be affected by desert dust mainly only in spring (Wu et al., 2009). Typically, the air masses coming from

south and south-west of Beijing moved slowly and spent much more time over industrialized regions,
resulting in high particulate matter mass concentrations. This here observed pattern is typical for Beijing,
and these connections between wind-direction and pollution levels in Beijing have been analyzed in
detail previously in Wehner et al. (2008).
**3.2 Particle number concentrations**
Fig. 4 shows the time series of the total number concentration of particles from 3 nm up to 2.5μm
($N_{total}$) and the number concentration of particles larger than 500 nm ($N_{>500nm}$), where $N_{total}$ varied from
$3*10^3$-$7*10^4$ cm$^{-3}$ and $N_{>500nm}$ varied from 10 to $4*10^3$ cm$^{-3}$. Obviously, in the atmosphere of Beijing
during the sampling period, small particles less than 500 nm account for a large faction of the total
particle number concentration, but during strong pollution events, also a large increase in $N_{>500nm}$ is seen.

Fig. 5(a) and Fig. 5(b) show INP number concentrations ($N_{INP}$) as a function of temperature for
INDA measurements. The lines are colour coded depending on the PM$_{2.5}$ mass concentration (Fig. 5(a))
and 12h-average $N_{>500nm}$ (Fig. 5(b)) during the respective filter sampling, where each line (30 in total)
represents an individual result of a filter. Exemplary measurement uncertainties are given in section 3 of
the appendix. All filter samples had INP that were active at -12.5°C and the highest freezing temperature
was observed to be -6°C. Overall, $N_{INP}$ varied from $10^{-3}$ to 1 L$^{-1}$. Already at a first glance, there is no
clear trend in $N_{INP}$ with PM$_{2.5}$ mass concentration and 12h-average $N_{>500nm}$, indicating that the dominant
pollutants of urban atmosphere may not significantly contribute to INPs active down to roughly -16°C
in an urban region.
To verify the results observed in INDA at lower temperatures, PM$_{2.5}$ collected by PTEF filters in
the same period were used for LINA which can test the ice nucleating properties of droplets down to
below -20°C. Washing particles off from the PTFE filters was more complete for some filters than for
others. This showed in differently large deviations in $N_{INP}$ from INDA and LINA measurements in the
overlapping temperature range, where results determined from INDA were always similar to or higher
than those from LINA as particle removal by washing the filters was frequently incomplete. It is
mentioned in Conen et al. (2012), that a quantitative extraction of particles from quartz fiber filters was
not possible without also extracting large amounts of quartz fibers. We tried to overcome this issue by

using PTFE filters, as degradation of the PTFE filter during washing does not occur due to the hydrophobic properties of the filter material. But we observed that not all particles were released into the water during the washing procedure (likely those collected deep within the filter), as filters frequently still looked greyish after washing, independent from the washing procedure (we experimented with different washing times up to 4 hours and with the use of an ultrasonic bath).

For our INDA measurements, punches of quartz filters were measured after they were immersed in water, representing the ice nucleating properties of all collected particles (Conen et al., 2012). However, as already mentioned above, $N_{INP}$ derived from LINA measurements were lower than those from INDA, due to particles that did not come off during washing. Based on our observations, we cannot recommend the use of sampling on PTFE filters followed by particle extraction in water. But we still decided to select those data from LINA measurements that showed the lowest deviation to the respective INDA results in the overlapping temperature range for use in this study. After calculating the deviation between INDA and LINA results, represented as the factor ($N_{INP}$ of INDA / $N_{INP}$ of LINA), ten LINA measurements from different days were selected to be used. For these measurements, the factor representing the deviation was in a range from 1.3 to 4.4. These data are shown in Fig. 5(c) and Fig. 5(d). The LINA data is represented by the dotted lines and the respective INDA data from the same sampling periods is represented by solid lines. In the temperature from -20°C to -25°C, results of LINA also show no clear trend in $N_{INP}$ with PM$_{2.5}$ mass concentration and 12h-average $N_{>500nm}$, even though a lower temperature has been involved, extending our statement that urban pollution might not contribute to INP down to -25°C.

**3.3 Correlation of $N_{INP}$ with PM$_{2.5}$, and BC mass concentration and particle number concentrations**

There have been many studies carried out in field and laboratory focusing on the ice nucleating properties of BC particles, however with inconclusive results. Some held the view that BC is not an efficient ice nucleation active species (Kamphus et al., 2010; Schill et al., 2016), whereas some described BC particles as possible INPs (Cozic et al., 2008; Cozic et al., 2007).

Here we selected $N_{INP}$ derived from INDA measurements at -16°C and plotted them against BC (Fig. 6 (a)), PM$_{2.5}$ mass concentrations (Fig. 6 (b)) and 12h-average values of $N_{total}$ (Fig. 6 (c)), $N_{>500nm}$ (Fig. 6 (d)), and $N_{INP}$ at -16°C derived from DeMott et al. (2010) (Fig.6 (e)) and DeMott et al. (2015) (Fig.6 (f)).

To determine the latter two, the 12h-averages of $N_{>500\,nm}$ shown in Fig. 3 were used, following
parameterizations suggested by DeMott et al. (2010, 2015). Linear fits are included in all panels of Fig.
6, and values for $R^2$ and p for these fits are shown in Table 1 Our results discussed in the following, based
on $N_{INP}$ at -16°C, are similarly valid for all other temperatures down to -25°C.

Fig. 6(a) to (f) show that there was no clear trend between $N_{INP}$ and any of the displayed parameters,

be it BC or $PM_{2.5}$ mass concentration or any of the 12h-average particle number concentrations. Also the
$R^2$ and p values given in Table 1 clearly show that there was no correlation between $N_{INP}$ and any of the
examined parameters. In the urban region of Beijing during winter, the INP could be assumed to be soot
or ash particles from traffic emissions, biomass burning and coal combustion, or to be dust particles
advected from the desert regions during prevailing northern and north-western wind, or to originate from
the biosphere. While mineral dust and biological particles are generally assumed to be the most abundant
INP in the atmosphere (Murray et al., 2012, Kanji et al., 2017), the role of particles from combustion,
i.e., of soot and ash particles, as INP is still controversial (Kanji et al., 2017). Our results indicate that
BC particles did not correlate with INP concentrations in the urban atmosphere. It is possible that the BC
particles emitted from coal burning, biomass burning, and traffic emissions are not ice active in the first
place, or that they underwent atmospheric aging processes (such as coagulation, condensation upon vapor,
and chemical reaction) resulting in more internally mixed particles after emission (Pöschl, 2005), which
might inactivate their potential to act as INP. In the atmosphere of Beijing, the aging timescale is much
shorter than in cleaner urban environments, which was shown in Peng et al. (2016). For example, to
achieve a complete morphology modification for BC particles in Beijing, the aging timescale was
estimated to be 2.3 h, compared to 9 h in Houston (Peng et al., 2016). $PM_{2.5}$ chemical composition
indicated that the BC particles may be aged and coated by secondarily formed chemical components
(SNA and other secondary organic materials) during the heavy haze episodes (Peng et al., 2016), thereby,
resulting in weakened heterogeneous ice nucleation activity of freshly emitted diesel soot particles
(Kulkarni et al., 2016).

However, if a possible coating was soluble, it would dissolve both during immersion freezing and

during our experiments and would not impede the ice activity of BC particles, unless it reacted chemically
with an ice active site. It has been observed that a coating did not impede the ice activity of mineral dust
particles coated with nitric acid in Sullivan et al. (2010) and coated with succinic acid or levoglucosan
in Wex et al. (2014).
Another study conducted in Ulaanbaatar in Mongolia, a city suffering from severe air pollution,
showed a low ice activity towards heterogeneous ice nucleation when the sulphur content of particles
was highest (Hasenkopf et al., 2016). It is interesting to note that we observe the opposite in our study,
i.e., the increase of $PM_{2.5}$ mass concentration and percentage of SNA in $PM_{2.5}$ during haze periods also
seem to have no negative impact on INP concentrations. Not only did increased BC mass concentrations
not increase the observed INP concentrations, but also were INP concentrations not particularly low
during pollution episodes. Furthermore, we conclude that the strong formation of secondary particulate
matter during haze days would not contribute to INP. In addition, there is no clear difference of ice
nucleation between day and night time samples.
The size distribution measurements show that the largest fraction of all particles occurred in the size
range below 500 nm. However, during the strongest pollution event towards the end of our measurement
period (Dec. 17 during daytime (1217D) till the night from Dec. 21 to Dec. 22 (1221N)), also $N_{>500nm}$
increased noticeably to much larger values than before. In general, also particles in this size range were
affected by the pollution, e.g., by an increase in size of pre-existing particles via atmospheric aging
processes (such as coagulation, condensation, chemical reaction) where particles advected from southern
industrial areas of Beijing might also contribute. This is at the base of the explanation why the
parameterizations for $N_{INP}$ by DeMott et al. (2010, 2015) were not able to describe the measured values,
as seen in Fig. 6 (e) and (f). Additionally, the time series of $N_{INP}$ at -16 °C, based on DeMott et al. (2010,
2015) and are shown as blue and green squares in Fig. 7, respectively. Also shown are values for $N_{INP}$ at
-16°C as measured by LINA (red circles), i.e., the same values used in Fig. 7. Mostly, the
parameterization by DeMott et al. (2015) yields larger values and a larger spread, compared to the
parameterization by DeMott et al. (2010), but naturally both follow the trends in $N_{>500nm}$. A correction
factor of 3, as suggested in DeMott et al. (2015), was not applied, as this would simply increase all
respective values by this factor, i.e., it will not change the results. Indeed, during the pollution phase, the
parameterizations overestimate the observed values by more than two orders of magnitude. But also
during clean phases, neither $N_{>500nm}$ nor the parameterizations by DeMott et al. (2010, 2015) correlate
with $N_{INP}$. Summarizing, this shows that pollution events not only did not add INP, but also that for the
aerosol observed during our study, a parameterization of $N_{INP}$ based on particles in the size range > 500
nm is not feasible. Interestingly, as will be shortly discussed in the next section, a much older
parameterization by Fletcher (1962) captures $N_{INP}$ as measured in this study rather surprisingly well, at

least within one order of magnitude (Fig. 8). In summary, during polluted days, the increase of BC concentration, secondary components (SNA) and other compounds contributing to $PM_{2.5}$, as well as particle concentrations have no impact on INP concentrations down to -25°C in the urban region we examined in our study. This means that anthropogenic pollution did not contribute to the INP concentration. But it also indicates that that anthropogenic pollution in Beijing did not deactivate the present INP, as polluted periods did not show particularly low INP concentrations, although aging and formation of secondary particulate matter typically are intense during times of strong pollution.

In addition to what we discussed above, also no correlation was observed between $N_{INP}$ and wind-speed, as can be seen by the respective values for $R^2$ and p given in Table 1. Fig. 9 indicates that there was also no correlation with wind-direction. The fact that we find no correlation with either wind-speed or wind-direction agrees with the desert regions towards the north-west not being efficient dust sources in winter, and are a hint that we may have observed average background INP concentrations in Beijing during our measurements.

Additionally, also no correlation was found between any of the water-soluble constituents that were analyzed with ion chromatography and INP concentrations. This is not too astounding, as INP make up only a small fraction of all particles, as can be seen when comparing number concentrations from Fig. 4 and Fig. 7, and hence they make up only a small fraction of the mass, likely too small to be detected. Furthermore, a number of different components might contribute to INP, e.g., biological INP that are generally ice active at higher temperatures (> -15°C) and mineral dusts which are ice active at lower temperatures, therefore one common tracer for INP might not be applicable. As far as K is concerned, which might be connected to K-feldspar containing mineral dust particles with high ice activity (Atkinson et al., 2013), we only analyzed the water soluble fraction, i.e., K related to feldspar would not have been analyzed. Moreover, K is also emitted by biomass burning and hence influenced by anthropogenic pollution. It remains to be seen if a simple correlation between chemical constituents of the atmospheric aerosol and INP concentrations can be established at all.

**3.4 Comparison with literature**

First, we compare our results with results of $N_{INP}$ derived from precipitation samples as collected in Petters and Wright (2015) as shown in Fig. 8. These literature data were collected in various locations

in North America and Europe, and none of these locations was one with strong anthropogenic pollution,
different from the sample location in the present study. The $N_{INP}$ in our study varied from $10^{-3}$-$10$ L$^{-1}$·air
at the temperature range of -10°C to -25°C. The data of this study (dark green and brownish lines) are
within the range of values given in Petters and Wright (2015), in the whole temperature range for which
INP concentrations were derived here. A comparison with Corbin et al. (2012) and Knopf et al. (2010),
who bothexamined INP also in urban air in Toronto and Mexico City, respectively, is not possible due
to different examined ice nucleation modes, and also because they only measured at -34°C (Corbin et al.,
2012), i.e., outside of the temperature range examined in this study, or only reported ice onset
temperatures (Knopf et al., 2010). But we want to point towards the fact that an older parameterization
based on Fletcher (1962), which has been used for large scale modelling, agrees well with our data (see
Fig. 8) down to -20°C. It should, however, also be pointed out that the occurring variability in the data
certainly cannot be captured by such a single line. But the increase in $N_{INP}$ towards lower temperatures
as parameterized in Fletcher (1962) is similar to that of our data, where it should also be said that this
parameterization is known to overestimate atmospheric observations at lower temperatures (roughly
below -25°C, see e.g., Meyers et al., 1992). A similar observation was recently described in Welti et al.
(2017), where down to -20°C the temperature trend of $N_{INP}$ derived from filter samples taken on the Cape
Verde islands also agreed well with the parameterisation by Fletcher (1962), while at lower temperatures,
the parameterization exceeded the measurements. In general, for the case of the Beijing air masses
examined in this study, both the range of $N_{INP}$ given in Petters and Wright (2015) as well as the
parameterization by Fletcher (1962) agree better with our measurements than the parameterizations by
DeMott et al. (2010, 2015).

All of this is again indicative for the fact that Beijing severe air pollution did not increase or decrease

INP concentrations above or below values typically observed in other, non-urban areas on the Earth, and
hence, that the background INP concentrations, at least down to -25°C might in general not be directly
anthropogenically influenced.

Measurements of $N_{INP}$ in China have been done as early as 1963 by You and Shi (1964), and a few

further studies listed in Table 2 have been carried out in recent years. Table 2 includes some campaigns
finished in different regions of China including mountains, plateaus and suburban districts with low PM$_{2.5}$
concentration and BC-containing particles. In contrast to these observations, our study shows $N_{INP}$
detected in an urban region during highly polluted days with complex particle sources. In our study,
immersion freezing was examined, while not all studies listed in Table 2 examined this ice nucleation
mode. But due to the scarcity of data, we include the results from all these studies in our discussion here.
Apparently, compared with results in Table 2, $N_{INP}$ determined for the urban site of this study (1 L$^{-1}$ Air
at -20°C) was on the lower end of reported values, which were up to roughly 20 L$^{-1}$ Air at -20°C for non-
dust events. Highest concentrations were observed for dust events with values up to 604 L$^{-1}$ · Air at -20°C
detected at a suburban site in Beijing, showing that INP from mineral dust contribute to the overall $N_{INP}$
already at this temperature (You et al., 2002). Despite the difference among methods and ice nucleating
modes, this again suggests that urban pollution aerosol particles might not be efficient immersion
freezing INP and that the ice nucleating ability of particles in urban aerosols might originate from the
non-urban background aerosol particles that are included in the urban aerosol, i.e., that INP observed in
urban environments might have the same sources among bioaerosols and dust particles as non-urban INP.
An additional contribution from urban biogenic or dust particles to the INP observed in this study cannot
be fully excluded, but the agreement between our data and rural data presented in literature (see Fig. 8
and Table 2) corroborates our assumption that atmospheric INP in general originate from non-urban
sources.
**4 Conclusions**
INP concentrations down to -25°C determined from PM$_{2.5}$ samples collected at an urban site of the
megacity Beijing, China, in winter were found to not be influenced by the highly variable amount of air
pollution, both in mass and particle number concentrations, that was present during the sampling period.
Therefore, we conclude that neither BC nor other pollutants contributed to INP, including secondarily
formed particulate mass. On the other hand, we also conclude that the present INP were not noticeably
deactivated during strong pollution events. Particle number concentrations for particles with diameters >
500nm were affected by pollution events, and INP concentrations did not correlate with these
concentrations. Therefore, as can be expected, parameterizations based on these concentrations (DeMott
et al., 2010, 2015) do not reproduce the INP concentrations under these extreme conditions and yield
values which are up to more than two orders of magnitude higher than the measured values. On the other
hand, INP concentrations were in the middle of the range reported for atmospheric, non-urban,
concentrations in Petters and Wright (2015), and on the lower end of reported values collected from
previous atmospheric observations in China, while they were much lower than observations during dust
events in China. From this, we conclude that INP concentrations might not be influenced directly by
anthropogenic activities, at least not down to roughly -25°C and maybe even below, and that particularly
natural mineral dust sources might effect INP concentrations observed in China. It should be noted that
ice nucleation observed at high freezing temperatures (particularly above -10°C, but maybe as low as -
20°C) is typically attributed to biogenic ice activity. But while identifying the nature of the INP detected
here is beyond the reach of our study, we assume that they originated from natural sources and not from
anthropogenic combustion sources. However, it should be kept in mind that an indirect anthropogenic
influence on INP concentrations is still possible due to land use changes and related changes in
atmospheric dust loadings as well as due to vegetation changes and related changes in the biosphere.





















**Appendix**

**1. Temperature calibration and background of INDA and LINA**

**1.1 Temperature calibration of INDA**

The bath of the thermostat was well mixed during the cooling cycle, and the cooling rate was at 1 K min$^{-1}$. PCR trays were immersed into the cooling liquid such that the water level in the tubes was below the level of the liquid in the thermostat. The temperature inside the tubes was determined before the experiments by putting a temperature sensor into a tube during cooling. This was repeated for tubes in several locations. This worked down to -7°C, below which the sensor induced freezing. In this temperature range, generally a small constant shift of 0.2 K was observed which was assumed to be overall valid and was incorporated in the data at all temperatures. A comparison of data obtained for suspensions of Snomax with previous work done at TROPOS with LACIS (Leipzig Aerosol Cloud Interaction Simulator) and within INUIT (Ice Nuclei Research Unit, (Wex et al., 2015)) showed good agreement down to the lowest temperature at which the experiments for the comparison were run, which was -16°C.

**1.2 Temperature calibration of LINA**

The temperature on the glass slide in LINA was obtained by feeding an air flow with a known dew point temperature through the instrument, while the instrument cooled down with 1 K min$^{-1}$, i.e., with the same freezing rate used in the experiments. The humidified air flow was obtained by mixing a dry air flow with an air flow that was humidified in a Nafion humidifier (Perma Pure MH-110-12S-4, Perma Pure, Toms River, New Jersey, USA) which was connected to a thermostat (HAAKE C25P, HAAKE GmbH, Karlsruhe, Germany) that kept the temperature in the humidifier at 10°C. By mixing the two air streams, dew point temperatures below 0°C were obtained. The dew point temperature was measured with a dew point mirror (Dew Prime I-S2, Edge Tech, Milford, Massachusetts, USA). The overall setup is based on the principle of a dew point mirror, i.e., the glass slide on the Peltier element in LINA started to fog when its temperature reached the dew point temperature adjusted in the air flow. Optical detection by the CCD camera was deployed similar to how it is used during measurements, i.e., taking a picture every 6 s. Subsequently detected greyscale images were compared to an image that was taken well before fogging began. Brightness differences between this original picture and the following pictures were taken and resulted in a S-shaped curve, reaching the maximum plateau once the glass slide was fogged over

completely. A fit was applied to the curve in order to find the temperature where 50% are fogged, which
was taken to represent the actual temperature. Using this principle, the temperature on the glass plate in
LINA was calibrated repeatedly at 5 different temperatures in the range from -2.3°C to -22.3°C. A
comparison of data obtained for suspensions of pollen washing water with previous work done at
TROPOS with LACIS (Augustin et al., 2013) showed good agreement down to the lowest temperature
at which the experiments for the comparison were run, which was -25°C.

**2. Background measurement of INDA and LINA**

In the background experiments of INDA, clean filters in distilled water froze from -17°C to -26°C,
while filters with atmospheric particles froze from -6°C to -22°C. The $f_{ice}$ of the clean filters was 5 to 14
times lower than that of atmospheric samples at the same temperature, showing a low impact. In LINA
measurements, the background of clean filters washed with distilled water was even lower, as droplets
started to freeze at -22°C. Figure A1 and A2 show the measured frozen fractions of the samples and the
background from pure water and the water with clean filters for both INDA and LINA, to corroborate
that the measurements were well separated from the background.

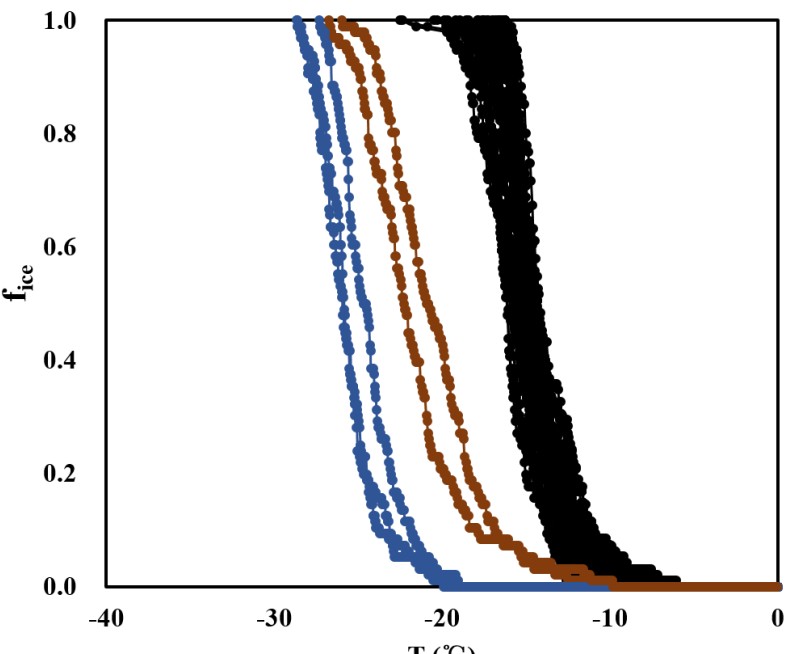


**Fig. A1 Frozen fractions determined from INDA (black lines), together with background signals determined**
**for pure water (blue lines) and for pure water containing punches of a clean filter (brown lines).**

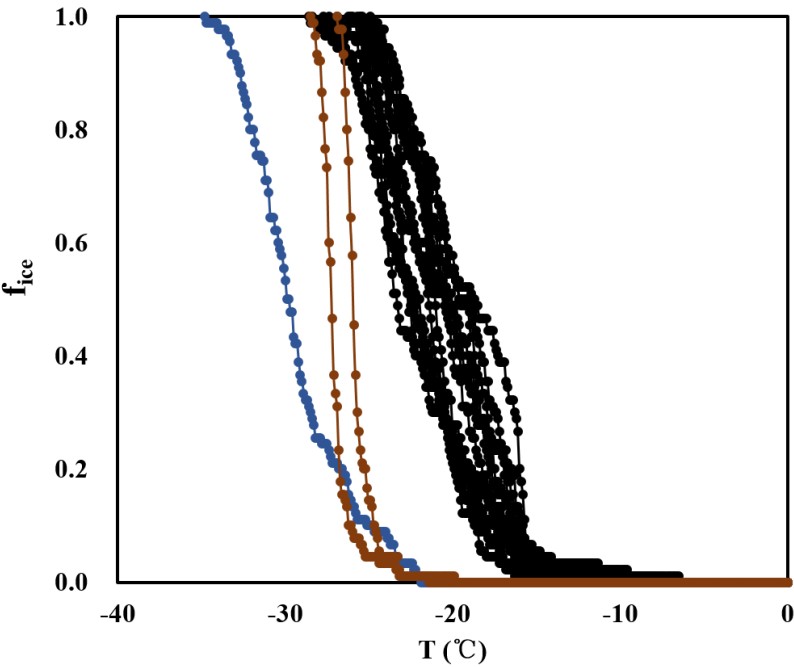

**Fig. A2 Frozen fractions determined from LINA, together with background signals determined for pure water**
**and for pure water in which a clean filter was put and washed, similar to the procedure for the samples.**

**3. Measurement uncertainty for INP measurements**


The highest and lowest freezing curved detected with INDA are shown exemplarily in Fig. A3

together with the measurement uncertainty. The derivation of the uncertainty was based on the fact that
at each temperature, all INP that are ice active at that or any higher temperature are Poission distributed
to the examined droplets. It followed a method described in Harrison et al. (2016). For LINA, no
uncertainties are given, as we know that washing off from the filters was incomplete, and the fraction of
particles that was retained on the filters cannot be determined. The largest deviation that we allowed
between LINA and INDA, i.e., a factor of 4.4 (see Sec. 3.2), is the base for the maximum uncertainty for
fice detected with LINA. For both, INDA and LINA, the temperature uncertainty is 0.5K.

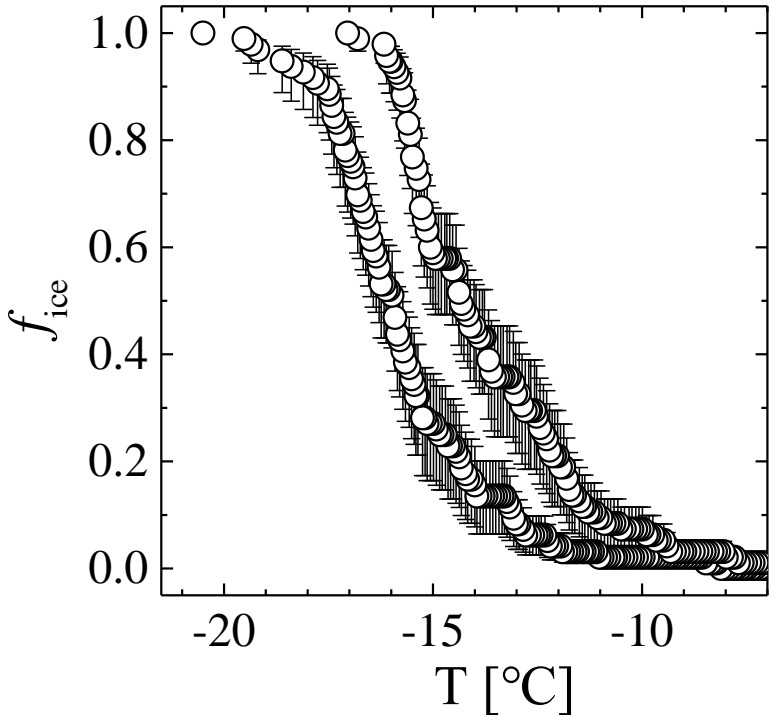

**Fig. A3. The highest and lowest freezing curved detected with INDA together with the measurement uncertainty.**

**Acknowledgments**

This work is supported by the following projects: National Natural Science Foundation of China (41475127, 41571130021) and Ministry of Science and Technology of the People's Republic of China (2016YFC0202801) and by the DFG funded Ice Nuclei Research Unit (INUIT, FOR 1525) (WE 4722/1-2) and Swedish Research Council (639-2013-6917).

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

**Table and Figures:**

Table 1 Coefficient of determination ($R^2$) and a measure for the statistical significance of the assumption of a
linear correlation (p) for the comparison of $N_{INP}$ at -16°C with the different parameters shown in Fig. 5.

| parameter | $R^2$ | p |
|---|---|---|
| (a)  BC concentration | 0.003 | 0.79 |
| (b)  PM$_{2.5}$ concentration | 0.006 | 0.71 |
| (c)  $N_{total}$ | 0.005 | 0.73 |
| (d)  $N_{>500nm}$ at -16°C | 0.008 | 0.67 |
| (e)  $N_{INP}$ at -16°C, based on DeMott et al. (2010) | 0.005 | 0.73 |
| (f)  $N_{INP}$ at -16°C, based on DeMott et al. (2015) | 0.007 | 0.67 |
| (g)  Wind speed | <0.001 | 0.99 |



**Table 2. Comparison of INP measurements in different regions of China, including $N_{INP}$ (i.e., INP number concentrations) and corresponding temperature**


| Sampling site | Citation | Sampling Date | Instruments | Temperature (°C) | Average INP (L$^{-1}$·Air) | Mode |
|---|---|---|---|---|---|---|
| **Huangshan** (mountain site) | (Jiang et al., 2015) | September-October,2012 | Vacauum water vapor diffusion chamber | -15~-23 | 0.27~7.02 | Deposition |
| **Huangshan** (mountain site) | (Jiang et al., 2014) | May-September,2011 | Mixing cloud chamber The static diffusion cloud chamber | -20 | 16.6 | Deposition/ Condensation |
| **Huangshan** (mountain site) | (Hang et al., 2014) | May-September,2011; September-October,2012 | Static vacuum water vapor diffusion cloud chamber | -20 | 18.74 | All modes |
| **Tianshan** (mountain site) | (Jiang et al., 2016) | 14-24 May, 2014 | Vacauum water vapor diffusion chamber; | -20 | 11(non-dust) Hundreds(dust) | Deposition |
| **Nanjing** (suburban site) | (Yang et al., 2012) | May-August,2011 | Mixing cloud chamber; The statistic diffusion chamber; | -20 | 20.11 | All modes |
| **Qing Hai** (plateau site) | (Shi et al., 2006) | 5-26 October, 2003 | The Bigger mixing cloud chamber | -15, -20, -25 | 23.3~85.4 | Deposition |
| **Beijing** (suburban site) | (You and Shi, 1964) | 18 March-30 April,1963 | Mixing cloud chamber | -20 | 3.9~4.8 | All modes |
| **Beijing** (suburban site) | (You et al., 2002) | 18 March-30 April,1995 | The Bigger mixing cloud chamber | -15, -20 | 21,78.9(non-dust) 604(dust) | All modes |
| **Beijing** (urban site) | This study | 27 November-22 December, 2016 | Ice Nucleation droplets Array | -10 ~ -28 | 0.001~10 | Immersion |






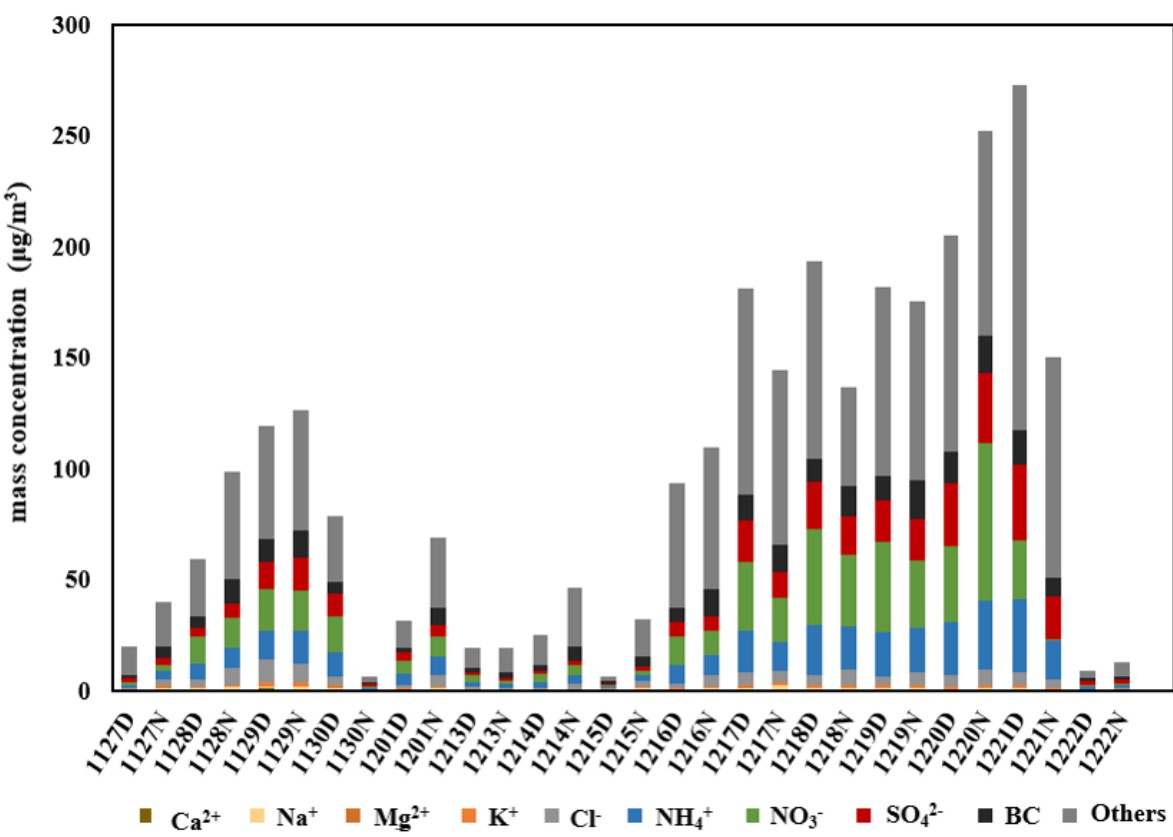

**Figure 1. The time series of PM$_{2.5}$ concentrations and chemical composition.    Data are shown for 15 different**
**days where the dates are indicated in the x-axis-labeling and "D" and "N" stand for daytime and nighttime,**
**respectively.**

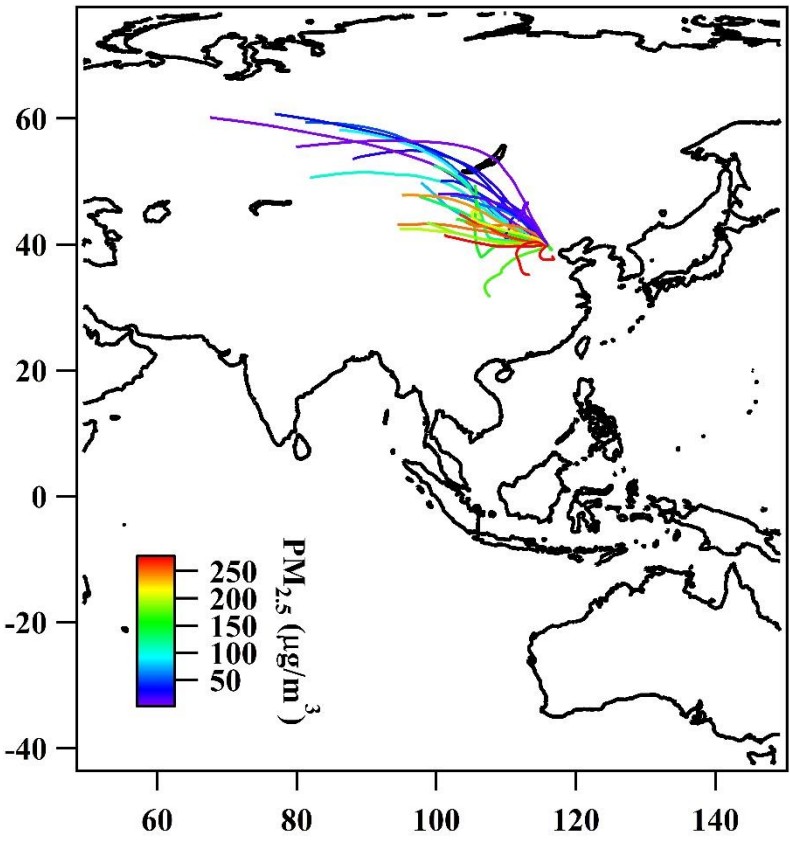

**Figure 2. The 2-day back-trajectories obtained by the NOAA HYSPLIT model colored-coded with respect to**
**PM$_{2.5}$ mass concentration determined by PTEF filter.**

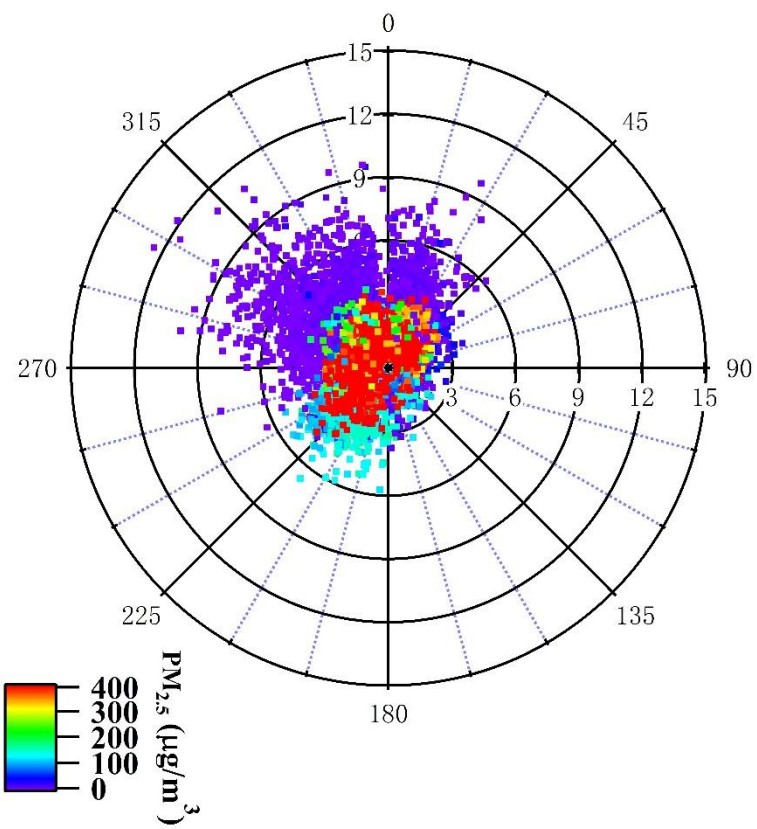

**Figure 3. Minutely recorded data for wind-direction and wind-speed colored-coded with respect to PM$_{2.5}$ mass**
**concentration.**

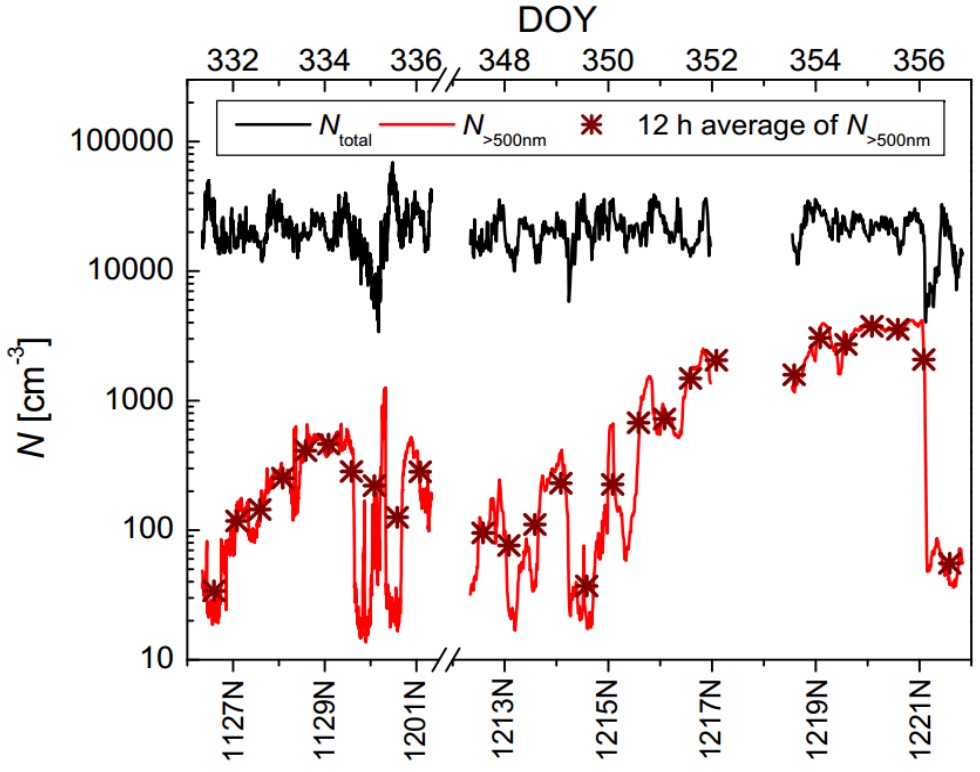


**Figure 4. The time series of $N_{total}$, $N_{>500nm}$ and 12-h average $N_{>500nm}$ at -16ºC.**

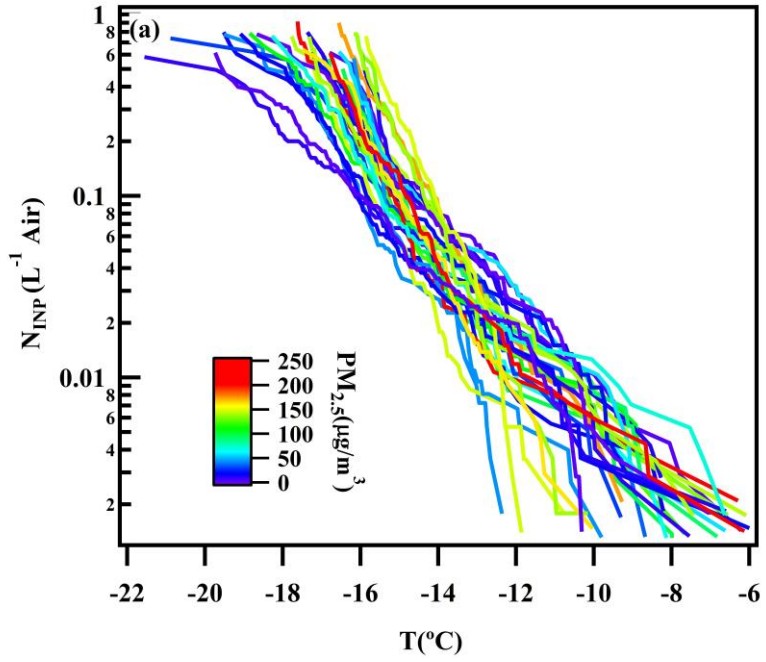


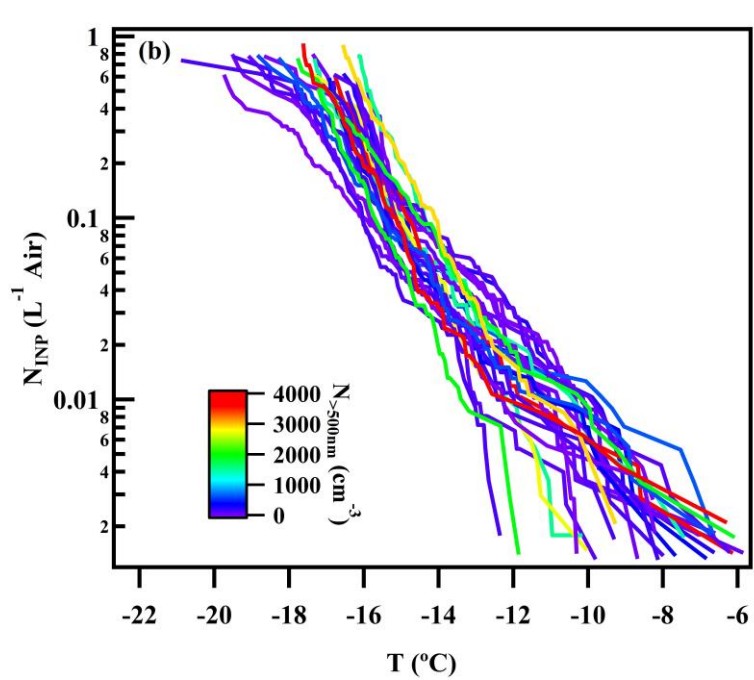


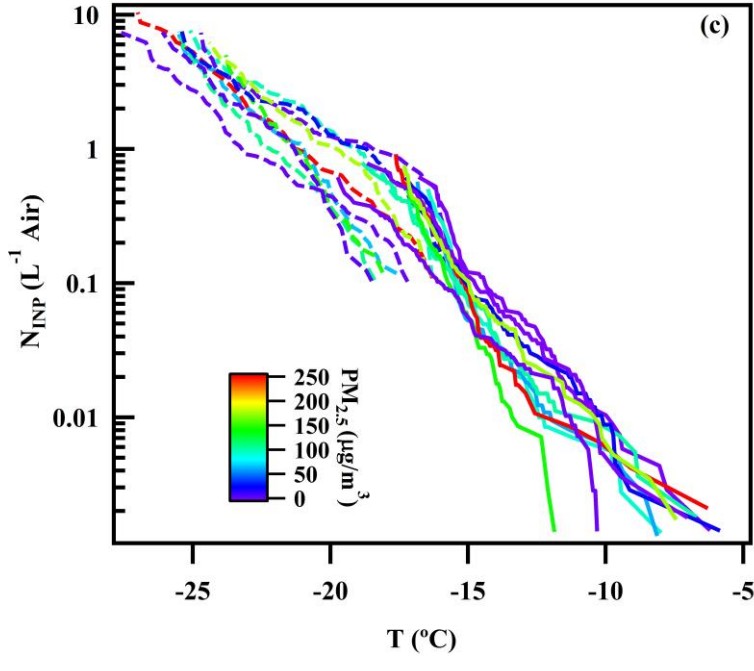


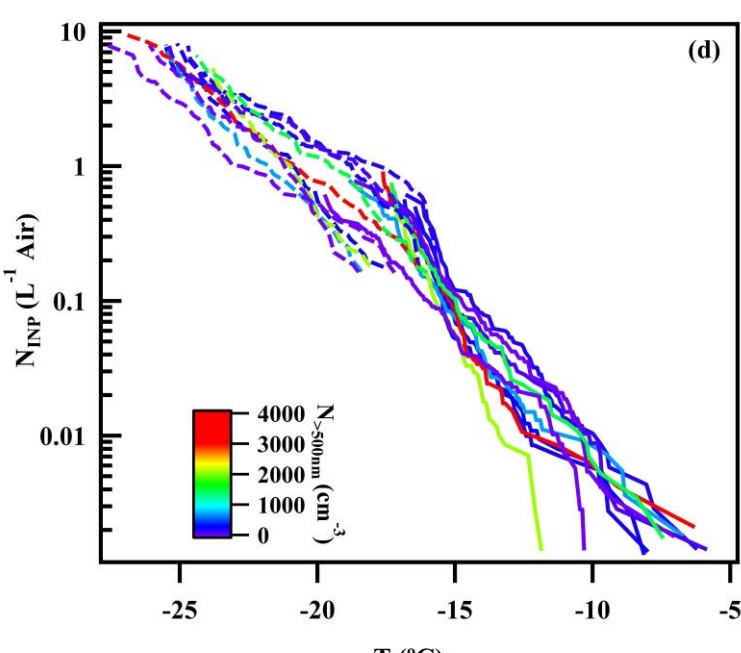



**Figure 5.** $N_{INP}$ **as function of temperature, panel (a) and (b) show INDA results coloured by PM2.5 mass**

**concentration and 12h-average $N_{>500nm}$, (c) and (d) for 10 comparable results of INDA and LINA coloured by**

**PM2.5 mass concentration and 12h-average $N_{>500nm}$, dotted lines represents LINA results while solid lines**

**represents INDA results.**


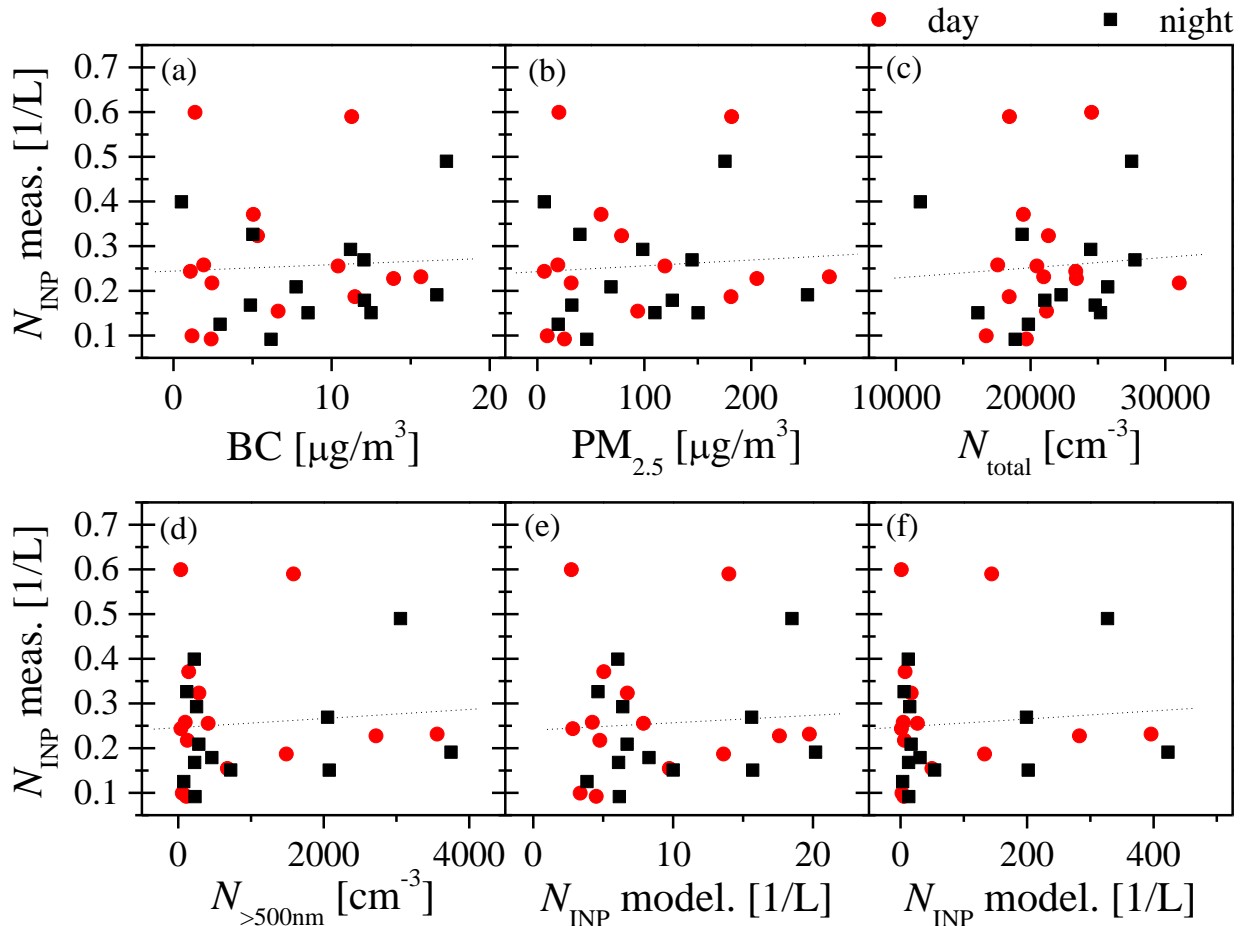



**Figure 6.** $N_{INP}$ at -16°C as function of mass concentrations of BC (a) and PM$_{2.5}$ (b), and of 12h-average values
of $N_{total}$ (c). Furthermore, we show $N_{>500nm}$ (d), and $N_{INP}$ at -16°C derived based on (DeMott et al., 2010) (e)
and DeMott et al. (2015) (f) for daytime (red round symbols) and nighttime (green square symbols) samples.

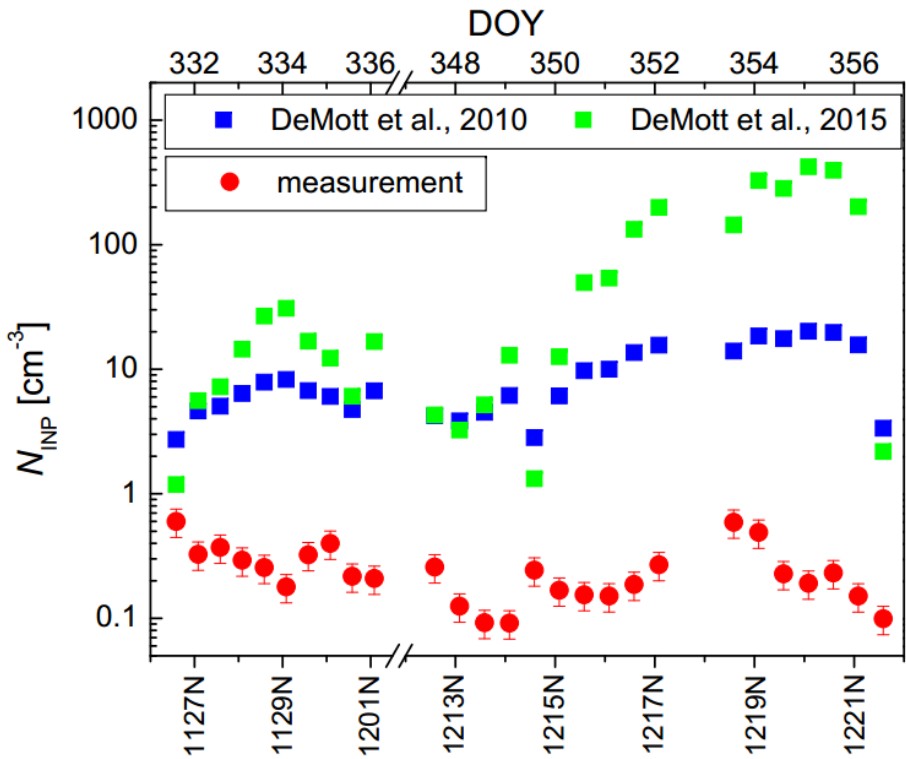


**Figure 7. The time series of measured $N_{INP}$ and $N_{INP}$ parameterized according to DeMott et al. (2010, 2015) at**

**-16ºC.**




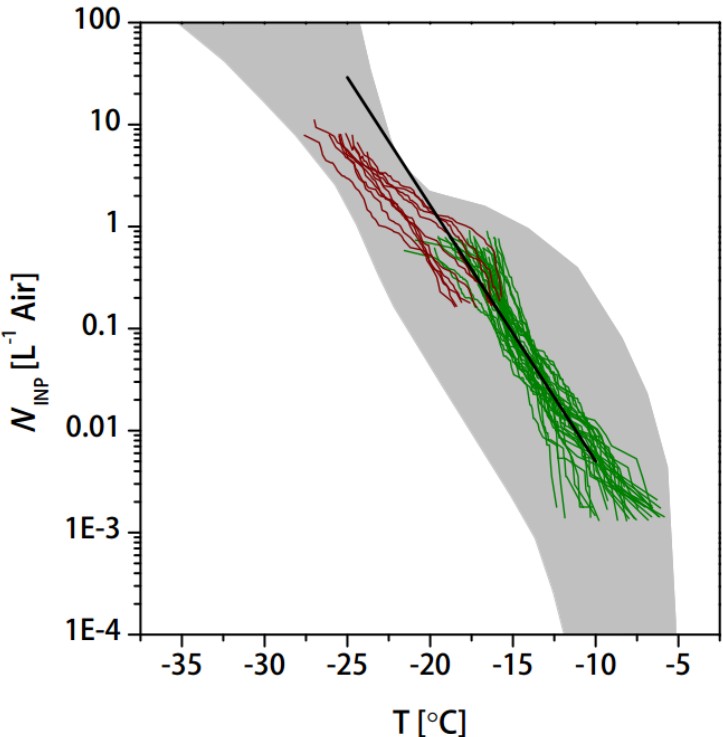


**Figure 8.** $N_{INP}$ **as derived from precipitation samples collected in Petters and Wright (2015) (grey area) and a**
**parameterization based on Fletcher (1962) (black line), together with our results (dark green and brownish**
**lines from INDA and LINA measurements, respectively).**


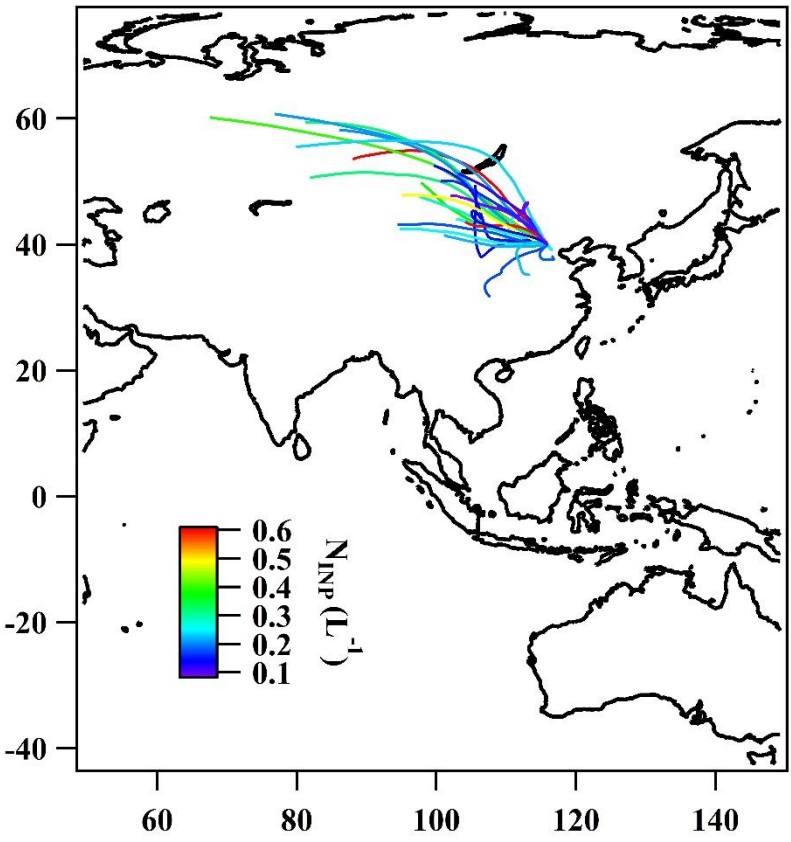


**Figure 9. The 2-day back-trajectories obtained by the NOAA HYSPLIT model colored-coded with respect**
**to INP concentration**

