# Peer review of "Ice nucleating particle concentrations unaffected by urban air pollution in Beijing, China"

_Atmospheric Chemistry and Physics, 2017_

## Referee Comment (RC1) · Anonymous Referee #1 · 10 Nov 2017

Review of "Ice nucleating particle concentrations unaffected by urban air pollution in Beijing, China" by Chen et al.

General Comment: This manuscript reports the ice nucleating abilities of urban aerosol particles from Beijing (China) using two different ice nuclei counters (i.e., LINA and INDA) during "clean" and heavy haze episodes. The authors did not find a major influence of the air pollution in Beijing on the ice nucleating particle (NP) concentrations as the INP levels did not correlate neither with PM2.5 nor with black carbon (BC) concentrations. Additionally, the predicted INP concentrations from the concentration of aerosol particles larger than 500 nm using DeMott et al. (2010) and DeMott et al. (2015) parametrizations did not correlate with the measured INP concentrations. The authors suggest that the INP concentrations in Beijing may have biogenic sources or

non-urban dust and that is why high levels of pollution did not increase the INP concentrations. This is one of the very few studies that measures the INP concentrations in an urban location, and it could be useful to help the ice nucleation community to understand the role that urban aerosol could play in ice cloud formation. However, the paper requires major corrections before it can be accepted for its publication in ACP.

Major comments: 1. Given that the "insignificant" influence of air pollution in the INP concentrations is not clearly supported I suggest to soften the tone of the Title. How about: "Ice nucleating particle concentrations under urban air pollution in Beijing, China"

2. The reviewer is surprised the authors completely ignored meteorology in this study. A detailed analysis of the meteorological variables and air masses is required to explain ambient observations, even in urban areas.

3. I am not sure if the comparison of the BC and INP concentration is completely fair given that the INP concentrations were obtained from particles collected on 8-h filters, while the BC data was obtained in-situ. Is it possible that the BC particles collected of the filters may have change their ice nucleating abilities during the 8-h period (i.e., aging, coagulation, oxidation, and coating)?

4. It is mentions that secondary particle formation did not contribute to the INP concentrations. However, it is complete unclear how secondary particles were measured or identified in this study.

5. The ice nucleation activation scans from the INDA and LINA are directly compared. However, given that the operational principle from both instruments is different, I am wondering if this is a fair comparison. Did the PTFE and quartz filters collect the same particles mass? Would it be necessary to normalize the INPs concentrations?

6. There are not uncertainties reported in this study at all. The correlations performed in this study do not present any statistical analysis.

7. I am not sure why the results from this study were compared with the Petters and Wright (2015) precipitation data. I would rather compare the present data with the results from Knopf et al. (2010) and Corbin et al. (2012) that were obtained in urban ambient air instead of precipitation samples.

8. The authors claim that the measured INPs are non-urban and they suggests that the sources of the INPs could be dust or bioparticles which are non-urban. Do the authors think that is it not possible to have urban dust and urban bioparticles?

9. The conclusions are not well supported by the shown data. They are mainly qualitative, in comes cases speculative, given the lack of meteorological analysis, and the non-detection of secondary organic particles, dust, and bioparticles.

Minor Comments: 1. It is unclear how BC was calculated/determined for the PM2.5 reported in Figure 1.

2. Why the measured INP concentration time series is not included in Figure 2?

3. There are several sentences and paragraphs that require a citation (e.g., Lines 34, 44, 250, 252, 264, 275, 280, and 352).

4. The introduction is quite disorganized. It jumps between bioparticles, dust, bioparticles, ash, soot, urban, soot and ash. I suggest to re-organize it and to focus on urban particles only. When introducing literature studies, make sure the ice nucleation modes are clearly stated.

5. Lines 85-87: Knopf et al. (2010) also performed immersion freezing experiments relevant to mixed-phase clouds.

6. Was the PM2.5 time series obtained from the particles collected on the PTFE or quartz filters?

7. Were the freezing experiments performed with the LINA recorded with pictures taken every 6 seconds? My experience is that droplets freeze very quick and if pictures are

taken every 6 seconds very important information can be missed.

8. It was said that particle removal by washing the filters was frequently incomplete. Can the authors indicate by how much? What percentage of the particles was not possible to be removed from the filters? This calculation is very important for the direct comparison of the LINA and INDA data.

9. Be consistent with the references format (lines 495, 505, 517, 553, 574, 604, 626, 629, 635, 642, 666, and 669).

10. Table 1. How is it possible to perform deposition ice nucleation at water saturation? Given that $S_i$ is higher than $S_w$, how is it possible to obtain conditions with $S_i= S_w$?

11. Figure 2. Are the INP concentrations in std L-1? Add here the measured INPs with their corresponding uncertainty.

12. Figure 3. Add all four panels to one single figure. I mean, one figure with 4 panels in one page.

13. Figure 4. Axis and symbols are too small. Add r2 and p-values.

14. Figure 5. I don't see the purpose of this figure given that Petters and Wrifgt (2015) study focused on precipitation samples.

---

## Referee Comment (RC2) · Anonymous Referee #2 · 11 Nov 2017

This paper presents measurements of ice nucleating particles in Beijing, China. The ice nucleation activity of particles sampled on filters was quantified using ice-nucleating droplet arrays, LINA and INDA. This information was supplemented by ion chromatography measurements of the filters and in-situ measurements of black carbon and particle size distributions. The authors find no correlation between filter-based INP concentrations and PM2.5 or black carbon measurements. As the authors correctly state, there are few measurements of ice nucleating particles in urban areas, particularly in China. This paper is therefore of interest to the community. I recommend the publication in ACP after the following concerns are addressed:

1. It would be useful to see a time series of INP concentrations alongside Figure 1.

2. On p. 10, lines 250-252, the authors mention difficulty in washing particles off

of PTFE filters and state that this procedure cannot be recommended in general. Why was the non-recommended technique used here? Can the authors provide an estimate of the uncertainty that this contributes to LINA measurements? Further, in line 252, the authors mention that they heave used a "subset" of PTFE filter LINA measurements. Which measurements were left off and why?

3. In Figure 4c, the x-axis label is "Ntotal at -16 degC". Should this be just the Ntotal average values from Figure 1? Why is the "at -16 degC" there?

4. Several (a and c, for example) of the plots in Figure 4 arguably show some weak correlations. Some statistical tests of significance would help to strengthen the authors' case.

5. Figure 4 only shows INP concentrations at -16 degC, were there any differences for other temperatures?

6. It would be useful to plot DeMott, et al. (2010 and 2015) parametrizations alongside the Fletcher (1962) parametrization in Figure 5.

7. The authors conclude that the INPs detected here are "background" INPs, likely originating from dust, based on some previous measurements in China, which show enhancements in ice nucleation during dust events. Since the calcium ion is used here as a tracer for dust, do the INP concentrations correlate with it? Do they correlate with any of the chemical constituents measured with ion chromatography?
* * *

---

## Author Comment (AC1) · 16 Jan 2018

**Answers to the report by anonymous Referee #1**

We thank Referee 1 for reviewing our manuscript and also for useful hints and suggestions. Below, comments from the referee are given in blue while our answers are given in black, with passages including new text given in italic. Additionally, the new text is marked yellow in the revised version of the manuscript.

General Comment:
This manuscript reports the ice nucleating abilities of urban aerosol
particles from Beijing (China) using two different ice nuclei counters (i.e., LINA and
INDA) during "clean" and heavy haze episodes. The authors did not find a major influence
of the air pollution in Beijing on the ice nucleating particle (NP) concentrations
as the INP levels did not correlate neither with PM2.5 nor with black carbon (BC) concentrations.
Additionally, the predicted INP concentrations from the concentration of
aerosol particles larger than 500 nm using DeMott et al. (2010) and DeMott et al.
(2015) parametrizations did not correlate with the measured INP concentrations. The
authors suggest that the INP concentrations in Beijing may have biogenic sources or
non-urban dust and that is why high levels of pollution did not increase the INP concentrations.
This is one of the very few studies that measures the INP concentrations
in an urban location, and it could be useful to help the ice nucleation community to
understand the role that urban aerosol could play in ice cloud formation. However, the
paper requires major corrections before it can be accepted for its publication in ACP.

Major Comments:
1. Given that the "insignificant" influence of air pollution in the
INP concentrations is not clearly supported I suggest to soften the tone of the Title.
How about: "Ice nucleating particle concentrations under urban air pollution in Beijing,
China"

Thank you for your suggestion, but we prefer not to change our title since we do think that our title is well supported by our results. Combined with the newly added information on meteorology, our laboratory results and different parameterizations, we show that the number concentration of INP has no correlation to some vital components of urban aerosol, where clean and heavily polluted days were examined. These components include, for example, BC concentration, particle number concentration and PM$_{2.5}$ mass concentration. We hope that you can agree to this for the new revised version of the manuscript.

2. The reviewer is surprised the authors completely ignored meteorology in this study.
A detailed analysis of the meteorological variables and air masses is required to explain
ambient observations, even in urban areas.

We added two plots showing trajectories and also wind direction and wind speed, to show the meteorological condition during the sampling period, together with the following text (line 234-245):

"*Additionally, Fig.2 shows 2-day back-trajectories obtained by the NOAA HYSPLIT model, with one trajectory related to each sampled filter, starting at the median sampling time of each filter. Fig. 3 shows minutely recorded data for wind-direction and wind–speed collected by (Met One 591) and (Met One 590) located on the same roof top as the aerosol sampling equipment. Both pictures are colored-coded with respect to PM$_{2.5}$ mass concentrations. The air masses that came from north or north-western directions were generally coincident with higher wind-speeds. They brought clean air with lower PM$_{2.5}$ mass concentrations. They did cross desert regions, however, Beijing was reported to be affected by desert dust in mainly only spring (Wu et al., 2009). Typically, the air masses coming from south and south-west of*

*Beijing moved slowly and spent much more time over industrialized regions, resulting in high particulate matter mass concentrations. This here observed pattern is typical for Beijing, and these connections between wind-direction and pollution levels in Beijing have been analyzed in detail previously in Wehner et al. (2008)."*

[Figure]

**Figure 2. The 2-day back-trajectories obtained by the NOAA HYSPLIT model colored-coded with respect to PM2.5 mass concentration determined by PTEF filter.**

[Figure]

**Figure 3. Minutely recorded data for wind-direction and wind-speed colored-coded with respect to PM$_{2.5}$ mass concentration.**

3. I am not sure if the comparison of the BC and INP concentration is completely fair given that the INP concentrations were obtained from particles collected on 8-h filters, while the BC data was obtained in-situ. Is it possible that the BC particles collected of the filters may have change their ice nucleating abilities during the 8-h period (i.e., aging, coagulation, oxidation, and coating)?

It is right that particles on the filters might age. But both BC measurements, in-situ and from filters, can be expected to yield the same results, as those properties of BC that would be detected from a filter measurement will not change with aging on the filters. So in general, the concern here would not be the comparison of 12-h-filter samples to averages of much more highly resolved aerosol data, but aging of BC such that a possible ice activity of the BC might be destroyed in general. Aging could happen due to loss of semi-volatile materials, such as ammonium nitrate, or through oxidation. A loss of semi-volatile material from BC is not to be expected. Concerning oxidation, during wintertime, atmospheric oxidants such as OH radicals and ozone typically have low concentrations. Ozone measurements done in parallel to our sampling showed that during nighttime, concentrations of ozone, one of the most important oxidants, was close to zero. During daytime, the ozone concentration was below 20 ppbv, indicating the oxidation capacity was very weak during our sampling in general. For coating, we assume you aim at the formation of SOA from the gas phase. These would be substances that would dissolve during INDA measurements when the sampled filters get in contact with the water, or when washing off the Teflon filters for LINA measurements. In general, during INDA measurements, water can be expected to surround all available surfaces, so particles being located close to each should not noticeably reduce the results, either.

Summarizing, we expect that the aging of BC particles collected on filters or other sampling artifacts concerning BC can be ignored for our study for INDA measurements. Concerning LINA (i.e., washing off the filters), see our answer to your minor comment 8.

4. It is mentions that secondary particle formation did not contribute to the INP concentrations. However, it is complete unclear how secondary particles were measured or identified in this study.

According to the chemical composition analysis in Figure 1, there was a notable increase in sulfate, nitrate and ammonium during times with high pollution. These substances are typically present in increased concentrations during secondary formation of particulate matter as shown in many studies, e.g., Zheng et al. (2016) and Guo et al. (2010). This formation of sulfate, nitrate and ammonium (SNA in short) is what we refer to as secondary formation of particulate matter (see the previous version of the manuscript, line 209). This and other related mentions might have been misleadingly formulated in the first version of our manuscript. We did the following changes:

formerly line 212, now line 221: We added "as it has previously been described in Guo et al. (2010) and Zheng et al. (2016). In this study, when we refer to secondarily formed particulate matter, this will always stand mainly for SNA and secondary organic substances."

formerly line 211, now line 220: We replaced "secondary transformation could be" by "*generation of secondary particulate mass is*"

formerly line 220, now line 231: We replaced "secondary and primary organic aerosols" by "*secondarily and primarily formed organic particulate matter*"

formerly line 292, now line 340: We replaced "secondary formation" by "*formation of secondary particulate matter*"

formerly line 316, now line 370: We replaced "secondary processes" by "*formation of secondary particulate matter*"

At the end of section 3.3, we explicitly state: "*Additionally, also no correlation was found between any of the water-soluble constituents that were analyzed with ion chromatography and INP concentrations.*" This means that also none of the components that formed secondary particulate mass contributed to INP.

5. The ice nucleation activation scans from the INDA and LINA are directly compared. However, given that the operational principle from both instruments is different, I am wondering if this is a fair comparison. Did the PTFE and quartz filters collect the same particles mass? Would it be necessary to normalize the INPs concentrations?

The samplers sampling filters for INDA and LINA were installed behind the same air inlet on a roof top, and the sampling length (12 hours) and times for filter changes on both samplers were the same. Therefore, it can be safely assumed that they sampled the same air mass. Please check what was already written in the manuscript (line 114 to 116 in the previous version, line 121 now): "Particles with an aerodynamic diameter less than or equal to 2.5 micro-meters (PM2.5) were collected on quartz fiber (Whatman, 1851-865) and PTFE filters (Whatman,7592-104) using a 4-channel sampler with 2.5μm impactors …"

If we would compare the measured frozen fractions obtained from INDA and LINA directly, indeed, the results could not be compared directly. However, by applying equation (2) (line 195 in the previously

submitted version, now line 204), a normalization is done with respect to the amount of air sampled per examined droplet. Therefore, the parameter we compare is the number of INP per volume of collected air for each of the methods. This makes data from the different methods directly comparable. Nothing changed.

6. There are not uncertainties reported in this study at all. The correlations performed
in this study do not present any statistical analysis.

Thank you for this hint. We prefer to not add error bars to the figures in the main text as this will make them unnecessarily messy and it would be difficult to see anything. Instead, we added a figure and text to the appendix and we also added error bars to the time series of $N_{INP}$ that is now shown in Fig. 7 (see our comment to your minor comment 2). Additionally we added data on the statistical analysis you were asking for.

In detail:

Uncertainties were added for measured frozen fractions in a new plot that appears in the appendix, together with descriptive text and the figure caption:

*"The highest and lowest freezing curved detected with INDA are shown exemplarily in Fig. A3 together with the measurement uncertainty. The derivation of the uncertainty was based on the fact that at each temperature, all INP that are ice active at that or any higher temperature are Poission distributed to the examined droplets. It followed a method described in Harrison et al. (2016). For LINA, no uncertainties are given, as we know that washing off from the filters was incomplete, and the fraction of particles that was retained on the filters cannot be determined. The largest deviation that we allowed between LINA and INDA, i.e., a factor of 4.4 (see Sec. 3.2), is the base for the maximum uncertainty for $f_{ice}$ detected with LINA. For both, INDA and LINA, the temperature uncertainty is 0.5K."*

[Figure]

**Fig. A3. The highest and lowest freezing curved detected with INDA together with the measurement uncertainty.**

As far as a statistical analysis is concerned, we added the following table, giving $R^2$ and p values for the 6 scatter-plots presented in what was Fig. 4 (now Fig. 6) and we added the respective lines to the figure (see

below), together with some text. It can clearly be seen by the values given in the newly added table, that there is no correlation.

The following text was added:
"*Linear fits are included in all panels of Fig. 6, and values for $R^2$ and p for these fits are shown in Table 1.*" (line 305)

"*Also the $R^2$ and p values given in Table 1 clearly show that there was no correlation between N_INP and any of the examined parameters.*" (line 309)

**Table 1: Coefficient of determination (R2) and a measure for the statistical significance of the assumption of a linear correlation (p) for the comparison of $N_{INP}$ at -16°C with the different parameters shown in Fig.6.**

| parameter | $R^2$ | p |
|---|---|---|
| (a) BC concentration | 0.003 | 0.79 |
| (b) $PM_{2.5}$ concentration | 0.006 | 0.71 |
| (c) $N_{total}$ | 0.005 | 0.73 |
| (d) $N_{>500nm}$ at -16°C | 0.008 | 0.67 |
| (e) $N_{INP}$ at -16°C, based on DeMott et al. (2010) | 0.005 | 0.73 |
| (f) $N_{INP}$ at -16°C, based on DeMott et al. (2015) | 0.007 | 0.67 |

[Figure]

**Figure 6.** $N_{INP}$ at -16°C as function of mass concentrations of BC (a) and PM$_{2.5}$ (b), and of 12h-average values of $N_{total}$ (c). Furthermore, we show $N_{>500nm}$ (d), and $N_{INP}$ at -16°C derived based on (DeMott et al., 2010) (e) and DeMott et al. (2015) (f) for daytime (red circles) and nighttime (black squares) samples.

7. I am not sure why the results from this study were compared with the Petters and Wright (2015) precipitation data. I would rather compare the present data with the results from Knopf et al. (2010) and Corbin et al. (2012) that were obtained in urban ambient air instead of precipitation samples.

In the study of Petters and Wright (2015), INP concentrations obtained per volume of precipitation are converted to INP concentrations per volume of air, i.e., to the same parameter we derive from our samples. Petters and Wright (2015) explain the uncertainties in their assumptions due to this conversion at great length, and we feel it is justified to use these data for the kind of comparison we are doing here. Also, Petters and Wright (2015) offer one of the largest compilations on atmospheric INP concentrations that we are aware off.

Concerning the publications from Knopf et al. (2010) and Corbin et al. (2012), as we already said in the introduction of the previously submitted version, their results were obtained for water sub-saturated conditions (see also a more detailed comment on Knopf et al., 2010 below, at your minor comment 5), i.e., no immersion freezing was examined, which impedes a direct comparison. Also, measurements in Corbin et al. (2012) were only done at -34°C, i.e., at a lower temperature than the range we examined in our study, while Knopf et al. (2010) only reports temperatures for ice nucleation onsets and no concentrations of INP, i.e., a comparison is not possible. This was added to the manuscript in the summarizing sentence given below (line 390 ff), while, for the here given multitude of reasons, nothing else was changed in our manuscript concerning this remark.

*"A comparison with Corbin et al. (2012) and Knopf et al. (2010), who both also examined INP in urban air in Toronto and Mexico City, respectively, is not possible due to different examined ice nucleation modes, and also because they only measured at -34°C (Corbin et al., 2012), i.e., outside of the temperature range examined in this study, or only reported ice onset temperatures (Knopf et al., 2010)."*

8. The authors claim that the measured INPs are non-urban and they suggest that the sources of the INPs could be dust or bioparticles which are non-urban. Do the authors think that is it not possible to have urban dust and urban bioparticles?

Indeed, there may be urban dust and also urban bioparticles. But these are no major contributor to the increase in PM2.5 mass concentrations during winter times – rather, it is well known that this increase is related to anthropogenic pollution. On the other hand, considering biogenic and dust particles, these particles emitted from urban areas will only contribute little to the overall atmospheric dust and biogenic particle load, as the non-urban sources are much more dominant for these types of particles. Therefore explicitly mentioning that dust and biogenic particles might also be emitted from urban sources does not really make sense. If there is, however, a specific passage in the text that you feel is miss-formulated, please tell us where this is exactly and why precisely you think this is wrong. For the time being, nothing was changed.

9. The conclusions are not well supported by the shown data. They are mainly qualitative, in comes cases speculative, given the lack of meteorological analysis, and the non-detection of secondary organic particles, dust, and bioparticles.

We do not agree with this rather simplified statement of the reviewer. We show that INP concentrations did neither correlate to concentrations of $PM_{2.5}$, BC, $N_{total}$ or $N_{>500nm}$ (and related to the latter also not to INP concentrations derived from parameterizations based on $N_{>500nm}$ taken from literature). It is correct that we do not know the nature of the INP we detect, but given their small total number (and hence mass), a chemical analysis to detect what they are is currently, and will be for quite some time, rather impossible, not only for us but for the community in general. Hence an indirect method as we show in the current work already adds a lot of understanding to connections between INP and different aerosol sources. Pollution in Beijing, at least for our data-set, did not add INP to the atmospheric aerosol.

We hope that by adding the trajectories and statistics on the correlations to the manuscript will help to convince the reviewer to step back from the statement he made here. As all respective changes were already discussed above, no additional changes were made.

Minor Comments:
1. It is unclear how BC was calculated/determined for the $PM_{2.5}$ reported in Figure 1.

The BC was measured by a multi-angle absorption photometer (5012, MAAP, Thermo Fisher Scientific, Waltham, MA, USA) which got the sampled air through an inlet with a 2.5μm cut-off. This had been included in the previous version of the manuscript. The instrument measures the absorption of particles collected on a filter with a time resolution of 5 min and automatically derives BC mass concentration from the measurement while accounting for multiple scattering occurring on the filter. The MAAP is a well known and often used instrument for the measurement of absorption coefficients and BC mass concentrations.

The respective retrieval of BC values is now added in line 143 ff:"*The instrument measures the absorption of particles collected on a filter with a time resolution of 5 min and automatically derives BC mass concentration from the measurement while accounting for multiple scattering occurring on the filter.*"

2. Why the measured INP concentration time series is not included in Figure 2?

We showed what was previously Figure 2 in a similar manner as we show Figure 1, i.e., to describe the general situation concerning the atmospheric aerosol. In this part of the text, derived INP number concentrations have not been shown, yet, and showing them here would twist the line of thought followed in the text. To be able to include the measured INP concentrations in the lower panel of the figure you refer to, here, we made that lower panel an extra figure (see below) which now appears as Figure 7. Related necessary changes were made in the text.

[Figure]

**Figure 7. The time series of measured $N_{INP}$ and $N_{INP}$ parameterized according to DeMott et al. (2010, 2015) at -16ºC.**

3. There are several sentences and paragraphs that require a citation (e.g., Lines 34, 44, 250, 252, 264, 275, 280, and 352).

We were somewhat astounded by the list given to us here, as one of these lines concerns a description we make in the text: "we decided to use a subset of the therewith obtained data. For our analysis, ten LINA measurements from different days were selected," (line 251-253 in the previously submitted manuscript). Another one already was a citation: "…, whereas some described BC particles as possible INPs (Cozic et al., 2008; Cozic et al., 2007)." (line 264 in the previously submitted manuscript). Where possible, we added citations where they were asked for:

34: We added the review by DeMott et al. (2010) here to support this statement: "This results in a significant impact on the cloud extent, lifetime, formation of precipitation, and radiative properties of clouds *(DeMott et al., 2010)*."

44: We added the review by Kanji et al. (2017) here to support this statement: "However, it has become obvious that many fundamental questions in this field are still unsolved *(Kanji et al., 2017)*."

250: See our answer to your minor comment 8.

275: We added: "*While mineral dust and biological particles are generally assumed to be the most abundant INP in the atmosphere (Murray et al., 2012, Kanji et al., 2017), the role of particles from combustion, i.e., of soot and ash particles, as INP is still controversial (Kanji et al., 2017)*."

280: We edited it and it now is: "*In the atmosphere of Beijing, the aging timescale is much shorter than in cleaner urban environments, which was shown in Peng et al. (2016).* For example, to achieve a complete morphology modification for BC particles in Beijing, the aging timescale was estimated to be 2.3 h *compared to 9 h in Houston* (Peng et al., 2016)."

352: The sentence referred to You et al. (2002) which was already given in Table 1 and is explicitly mentioned in the text now as well.

4. The introduction is quite disorganized. It jumps between bioparticles, dust, bioparticles, ash, soot, urban, soot and ash. I suggest to re-organize it and to focus on urban particles only. When introducing literature studies, make sure the ice nucleation modes are clearly stated.

Focussing only on urban particles makes no sense, as we show in this work that urban pollution particles do not contribute to INP. Indeed, as you remarked elsewhere, the urban environment can add mineral dust and bioparticles. But the fact that our derived INP concentrations are well within those reported in Petters & Wright (2015), where the latter were derived from non-urban environments, suggests that we are measuring a typical mid-latitude continental background, where non-urban sources might be the largest contributors. Hence it makes sense to at least mention all these possible types of INP particles.

We have reorganized the introduction, now shown at line 31 ff. Please see the revised version of the manuscript. New text is marked in yellow, but parts of the text that were simply shifted to improve the flow of the text are not marked particularly.

5. Lines 85-87: Knopf et al. (2010) also performed immersion freezing experiments relevant to mixed-phase clouds.

We are a bit confused by your statement, as Fig. 1 in Knopf et al. (2010), which presents all the data obtained for ice nucleation in that study, shows that data was not obtained at water saturation, besides for mainly the background measurements (called "H2O uptake substrate") and a tiny fraction of one bar representing some data that barely touches the water saturation line. Indeed, there is data in there that is called "immersion mode", however, as these data were generally obtained at relative humidities well below saturation (wrt. liquid water), this is not really immersion freezing. As one of us showed in a previous publication (Wex et al., 2014), for the type of freezing called "immersion freezing" in Knopf et al. (2010) an additional freezing point depression would have to be considered. Also, results on INP in Knopf et al. (2010) are only presented in terms of temperatures for the onset for ice nucleation, and the

text only mention cirrus clouds. The relevance of this paper for mixed-phase clouds eludes us. Nothing changed.

6. Was the PM2.5 time series obtained from the particles collected on the PTFE or quartz filters?

We point you towards the previously submitted manuscript where we already said: "Two PTEF filters were always sampled in parallel, and while one was used for INP analysis, the other was selected for the total mass and water-soluble ion analysis." (Previously line 139-140, now line 149-150.)

7. Were the freezing experiments performed with the LINA recorded with pictures taken every 6 seconds? My experience is that droplets freeze very quick and if pictures are taken every 6 seconds very important information can be missed.

Yes, pictures were taken every 6 seconds. And yes, droplets freeze very quickly. But we do not have to detect the exact second at which a droplet freezes, and once it is frozen, it stays frozen during the experiment. For a cooling rate of 1 K/min, data-points can "only" be given for a temperature resolution of 0.1 K, which, however, is still a high temperature resolution. Each time a picture is taken, the cumulated number concentration of INP that are ice active at the respective temperature (that is currently effectively valid) or above can be derived (by counting all frozen droplets).

This is a procedure that has been used in the past and currently has seen a revival, as described e.g. in the literature by Budke & Koop (2015) and Conen et al. (2012), on which our set-ups and methods are based, as we describe in the manuscript.

8. It was said that particle removal by washing the filters was frequently incomplete. Can the authors indicate by how much? What percentage of the particles was not possible to be removed from the filters? This calculation is very important for the direct comparison of the LINA and INDA data.

As we said in the previously submitted version of our manuscript, of the 30 examined filter sets, for ten of the filters for which washing off was done the deviation factor between INP concentrations from INDA and LINA were between 1.3 and 4.4 (INDA was always higher), and only results from the analysis with LINA for these ten filters are shown in the manuscript. This had been said in lines 252 to 254 in the previous version.

We edited the respective paragraph and added some additional information, so this whole passage now is as follows (with new text in italic) (see line 270 ff):
"Washing particles off from the PTFE filters was more complete for some filters than for others. This showed in differently large deviations in $N_{INP}$ from INDA and LINA measurements in the overlapping temperature range, where results determined from INDA were always similar to or higher than those from LINA, as particle removal by washing the filters was frequently incomplete. *It is mentioned in Conen et al. (2012), that a quantitative extraction of particles from quartz fiber filters was not possible without also extracting large amounts of quartz fibers. We tried to overcome this issue by using PTFE filters, as degradation of the PTFE filter during washing does not occur due to the hydrophobic properties of the filter material. But we observed that not all particles were released into the water during the washing procedure (likely those collected deep within the filter), as filters frequently still looked greyish after washing, independent from the washing procedure (we experimented with different washing times up to 4 hours and with the use of an ultrasonic bath). For our INDA measurements, punches of quartz filters were*

*measured after they were immersed in water, representing the ice nucleating properties of all collected particles (Conen et al., 2012). However, as already mentioned above, $N_{INP}$ derived from LINA measurements were lower than those from INDA, due to particles that did not come off during washing. Based on our observations, we cannot recommend the use of sampling on PTFE filters followed by particle extraction in water. But we still decided to select those data from LINA measurements that showed the lowest deviation to the respective INDA results in the overlapping temperature range for use in this study. After calculating the deviation between INDA and LINA results, represented as the factor ($N_{INP}$ of INDA / $N_{INP}$ of LINA), ten LINA measurements from different days were selected to be used. For these measurements, the factor representing the deviation was in a range from 1.3 to 4.4."*

9. Be consistent with the references format (lines 495, 505, 517, 553, 574, 604, 626, 629, 635, 642, 666, and 669).

Thank you for your careful reading, we corrected these citations.

10. Table 1. (Now Table 2!) How is it possible to perform deposition ice nucleation at water saturation? Given that S_i is higher than S_w, how is it possible to obtain conditions with S_i= S_w?

It is difficult to convey all the information from the literature cited here in a simple table. This caused the information on S_i and S_w to be somewhat messy. We decided to delete this information, as it does not add any additional value on top of the freezing mode that was examined in the different cited papers and that is given in the table, anyway.

11. Figure 2. (Now Figure 4) Are the INP concentrations in std L-1? Add here the measured INPs with their corresponding uncertainty.

Yes, the volume we used was already given as standard volume. This is explicitly mentioned in the text now.

As mentioned above at our answer to your minor comment 2, the data was added, but for the flow of the text, the lower panel to which the data was added is now an extra figure which now appears as Figure 7. (See our answer to your minor comment 2, above.)

As mentioned above at our answer to your major comment 6, a new figure and related text was added in the appendix.

12. Figure 3. (Now Figure 5) Add all four panels to one single figure. I mean, one figure with 4 panels in one page.

This is a topic for setting the final version of the manuscript. Nothing changed.

13. Figure 4. (Now Figure 6) Axis and symbols are too small. Add r2 and p-values.

The figure was edited, script has been enlarged, and a table with R2 and p values has been added. See our answer to your major comment 6.

14. Figure 5. (Now Figure 8) I don't see the purpose of this figure given that Petters and Wrifgt (2015) study focused on precipitation samples.

As we already argued in our comment to your major comment 7, in the study of Petters and Wright (2015), INP concentrations obtained per volume of precipitation are converted to INP concentrations per volume of air, i.e., to the same parameter we derive from our samples. Petters and Wright (2015) explain the uncertainties in their assumptions due to this conversion at great length, and we feel it is justified to use these data for the kind of comparison we are doing here. Also, Petters and Wright (2015) offer one of the largest compilations on atmospheric INP concentrations that we are aware off. We feel that it is an important information that INP number concentrations in such a strongly polluted location as Beijing city in November and December did not exceed the respective concentrations measured in more rural environments.

Literature:

Conen, F., Henne, S., Morris, C. E., and Alewell, C.: Atmospheric ice nucleators active>=12°C can be quantified on PM10 filters, Atmos. Meas. Tech., 5, 321-327, doi:10.5194/amt-5-321-2012, 2012.

Corbin, J. C., Rehbein, P. J. G., Evans, G. J., and Abbatt, J. P. D.: Combustion particles as ice nuclei in an urban environment: Evidence from single-particle mass spectrometry, Atmos. Environ., 51, 286-292, doi:10.1016/j.atmosenv.2012.01.007, 2012.

Cozic, J., Verheggen, B., Mertes, S., Connolly, P., Bower, K., Petzold, A., Baltensperger, U., and Weingartner, E.: Scavenging of black carbon in mixed phase clouds at the high alpine site Jungfraujoch, Atmos. Chem. Phys., 7, 1797-1807, doi:10.5194/acp-7-1797-2007, 2007.

Cozic, J., Mertes, S., Verheggen, B., Cziczo, D. J., Gallavardin, S. J., Walter, S., Baltensperger, U., and Weingartner, E.: Black carbon enrichment in atmospheric ice particle residuals observed in lower tropospheric mixed phase clouds, J. Geophys. Res., 113, doi:10.1029/2007jd009266, 2008.

DeMott, P. J., Prenni, A. J., Liu, X., Kreidenweis, S. M., Petters, M. D., Twohy, C. H., Richardson, M. S., Eidhammer, T., and Rogers, D. C.: Predicting global atmospheric ice nuclei distributions and their impacts on climate, Proc. Natl. Acad. Sci., 107, 11217-11222, doi:10.1073/pnas.0910818107, 2010.

Guo, S., Hu, M., Wang, Z. B., Slanina, J., and Zhao, Y. L.: Size-resolved aerosol water-soluble ionic compositions in the summer of Beijing: implication of regional secondary formation, Atmos. Chem. Phys., 10, 947-959, doi:10.5194/acp-10-947-2010, 2010.

Harrison, A. D., T. F. Whale, M. A. Carpenter, M. A. Holden, L. Neve, D. O'Sullivan, J. V. Temprado, and B. J. Murray: Not all feldspars are equal: a survey of ice nucleating properties across the feldspar group of minerals, Atmos. Chem. Phys., 16, 10927–10940, doi:10.5194/acp-16-10927-2016, 2016.

Knopf, D. A., Wang, B., Laskin, A., Moffet, R. C., and Gilles, M. K.: Heterogeneous nucleation of ice on anthropogenic organic particles collected in Mexico City, Geophys. Res. Lett., 37, doi:10.1029/2010GL043362, 2010.

Petters, M. D., and Wright, T. P.: Revisiting ice nucleation from precipitation samples, Geophys. Res. Lett., 42, 8758-8766, doi:10.1002/2015GL065733, 2015.

Peng J, Hu M, Guo S, et al. Markedly enhanced absorption and direct radiative forcing of black carbon under polluted urban environments, Proc Natl Acad Sci U.S.A., 2016, 113(16):4266, doi: 10.1073/pnas.1602310113, 2016.

Wex, H., P. J. DeMott, Y. Tobo, S. Hartmann, M. Rösch, T. Clauss, L. Tomsche, D. Niedermeier, and F. Stratmann: Kaolinite particles as ice nuclei: learning from the use of different kaolinite samples and different coatings, Atmos. Chem. Phys., 14, 5529-5546, doi:10.5194/acp-14-5529-2014, 2014.

Wu, Z. J., Cheng, Y. F., Hu, M., Wehner, B., Sugimoto, N., and Wiedensohler, A.: Dust events in Beijing, China (2004–2006): comparison of ground-based measurements with columnar integrated observations, Atmos. Chem. Phys., 9, 6915-6932, doi:10.5194/acp-9-6915-2009, 2009.

Wehner, B., Birmili, W., Ditas, F., Wu, Z., Hu, M., Liu, X., Mao, J., Sugimoto, N., and Wiedensohler, A.: Relationships between submicrometer particulate air pollution and air mass history in Beijing, China, 2004–2006, Atmos. Chem. Phys., 8, 6155-6168, doi:10.5194/acp-8-6155-2008, 2008.

Zheng, J., Hu, M., Peng, J., Wu, Z., Kumar, P., Li, M., Wang, Y., and Guo, S.: Spatial distributions and chemical properties of PM2.5 based on 21 field campaigns at 17 sites in China, Chemosphere,159, 480-487, doi:10.1016/j.chemosphere.2016.06.032, 2016

---

## Author Comment (AC2) · 16 Jan 2018

**Answers to the report by anonymous Referee #2**

We thank Referee 2 for reviewing our manuscript and also for useful hints and suggestions. Below, comments from the referee are given in blue while our answers are given in black, with passages including new text given in italic. Additionally, the new text is marked yellow in the revised version of the manuscript.

This paper presents measurements of ice nucleating particles in Beijing, China. The ice nucleation activity of particles sampled on filters was quantified using ice-nucleating droplet arrays, LINA and INDA. This information was supplemented by ion chromatography measurements of the filters and in-situ measurements of black carbon and particle size distributions. The authors find no correlation between filter-based INP concentrations and PM2.5 or black carbon measurements. As the authors correctly state, there are few measurements of ice nucleating particles in urban areas, particularly in China. This paper is therefore of interest to the community. I recommend the publication in ACP after the following concerns are addressed:

1. It would be useful to see a time series of INP concentrations alongside Figure 1.

We included such a time series in the lower panel of what was Figure 2 (now Fig. 7), and, for the flow of the text, made that lower panel an extra figure which now appears as Figure 7. Related necessary changes were made in the text.

[Figure]

**Figure 7. The time series of measured $N_{INP}$ and $N_{INP}$ parameterized according to DeMott et al. (2010, 2015) at -16ºC.**

2. On p. 10, lines 250-252, the authors mention difficulty in washing particles off of PTFE filters and state that this procedure cannot be recommended in general. Why

was the non-recommended technique used here? Can the authors provide an estimate of the uncertainty that this contributes to LINA measurements? Further, in line 252, the authors mention that they heave used a "subset" of PTFE filter LINA measurements. Which measurements were left off and why?

While doing the sampling, we had planned to focus on the quartz fiber filters, knowing that this will yield results for a restricted temperature range, only. But as we had a four-channel sampler, we thought we could try to sample on additional filter material. Polycarbonate filters did not work out, as they cause such a large pressure drop and the pump that was available was not sufficiently strong to collect a reasonable air volume. Hence the PTFE filters were used. We were not sure if these filters would work, but tried, nevertheless. It turned out that indeed, due to the fibers, particles stick in them even after washing (filters sometimes were still colored after the washing), but we decided to use at least results from those filters for which, in the temperature range where data was obtained from both filters, results were not more than a factor of 4.4 lower for LINA than for INDA data, which left data from ten filters analyzed with LINA.

We edited the text concerning this and added some additional information, so this whole passage now is as follows (with new text in italic) (see line 270):

"Washing particles off from the PTFE filters was more complete for some filters than for others. This showed in differently large deviations in $N_{INP}$ from INDA and LINA measurements in the overlapping temperature range, where results determined from INDA were always similar to or higher than those from LINA, as particle removal by washing the filters was frequently incomplete. *It is mentioned in Conen et al. (2012), that a quantitative extraction of particles from quartz fiber filters was not possible without also extracting large amounts of quartz fibers. We tried to overcome this issue by using PTFE filters, as degradation of the PTFE filter during washing does not occur due to the hydrophobic properties of the filter material. But we observed that not all particles were released into the water during the washing procedure (likely those collected deep within the filter), as filters frequently still looked greyish after washing, independent from the washing procedure (we experimented with different washing times up to 4 hours and with the use of an ultrasonic bath). For our INDA measurements, punches of quartz filters were measured after they were immersed in water, representing the ice nucleating properties of all collected particles (Conen et al., 2012). However, as already mentioned above, $N_{INP}$ derived from LINA measurements were lower than those from INDA, due to particles that did not come off during washing. Based on our observations, we cannot recommend the use of sampling on PTFE filters followed by particle extraction in water. But we still decided to select those data from LINA measurements that showed the lowest deviation to the respective INDA results in the overlapping temperature range for use in this study. After calculating the deviation between INDA and LINA results, represented as the factor ($N_{INP}$ of INDA / $N_{INP}$ of LINA), ten LINA measurements from different days were selected to be used. For these measurements, the factor representing the deviation was in a range from 1.3 to 4.4."*

3. In Figure 4c, the x-axis label is "Ntotal at -16 degC". Should this be just the Ntotal average values from Figure 1? Why is the "at -16 degC" there?

You are right, thanks for the hint. Corrected!

4. Several (a and c, for example) of the plots in Figure 4 arguably show some weak

correlations. Some statistical tests of significance would help to strengthen the authors' case.

We added the following table giving $R^2$ and p values for the 6 scatter-plots presented in in what was Fig. 4 (now Fig. 6) and added the respective lines to the figure (see below), together with some text. It can clearly be seen by the values given in the newly added table, that there is no correlation.

The following text was added:
"*Linear fits are included in all panels of Fig. 6, and values for $R^2$ and p for these fits are shown in Table 1.*" (line 305)

"*Also the $R^2$ and p values given in Table 1 clearly show that there was no correlation between N_INP and any of the examined parameters.*" (line 309)

**Table 1 Coefficient of determination ($R^2$) and a measure for the statistical significance of the assumption of a linear correlation (p) for the comparison of $N_{INP}$ at -16°C with the different parameters shown in Fig. 6.**

| parameter | $R^2$ | p |
|---|---|---|
| (a) BC concentration | 0.003 | 0.79 |
| (b) $PM_{2.5}$ concentration | 0.006 | 0.71 |
| (c) $N_{total}$ | 0.005 | 0.73 |
| (d) $N_{>500nm}$ at -16°C | 0.008 | 0.67 |
| (e) $N_{INP}$ at -16°C, based on DeMott et al. (2010) | 0.005 | 0.73 |
| (f) $N_{INP}$ at -16°C, based on DeMott et al. (2015) | 0.007 | 0.67 |

[Figure]

**Figure 6.** $N_{INP}$ at -16°C as function of mass concentrations of BC (a) and PM$_{2.5}$ (b), and of 12h-average values of $N_{total}$ (c). Furthermore, we show $N_{>500nm}$ (d), and $N_{INP}$ at -16°C derived based on (DeMott et al., 2010) (e) and DeMott et al. (2015) (f) for daytime (red round symbols) and nighttime (green square symbols) samples.

5. Figure 4 (Now Figure 6) only shows INP concentrations at -16 degC, were there any differences for other temperatures?

No, there were no correlations at other temperatures, either (which can already be seen by the freezing curves not crossing much). This had been said before in the manuscript (previously line 268, now line 306), therefore nothing was changed: "Our results discussed in the following, based on $N_{INP}$ at -16°C, are similarly valid for all other temperatures down to -25°C."

6. It would be useful to plot DeMott, et al. (2010 and 2015) parametrizations alongside the Fletcher (1962) parametrization in Figure 5.

The DeMott parameterizations were added to the respective figure as shown below. However, we do prefer to not add this to the main manuscript as we do not want to stress it too much that these parameterizations do not fit, as this is caused by the pollution aerosol, which also included pollution particles larger than 500 nm, and not by the parameterizations. It does not say anything about the applicability of the parameterizations. Should you be of a different opinion, please let us know.

[Figure]

7. The authors conclude that the INPs detected here are "background" INPs, likely originating from dust, based on some previous measurements in China, which show enhancements in ice nucleation during dust events. Since the calcium ion is used here as a tracer for dust, do the INP concentrations correlate with it? Do they correlate with any of the chemical constituents measured with ion chromatography?

We did not find a correlation with any single component we looked at, please see the comparison with $Ca^{2+}$ and $K^+$ as examples below (this is not included in the new version of the manuscript, we only show it to you here). It has been assumed that the feldspar fraction (and there maybe only the K-feldspar fraction) of dust may be responsible for the ice nucleation activity of atmospheric mineral dust, therefore it is not too astounding that no correlation with $Ca^{2+}$ was found (see both figures below). As for $K^+$, this is also emitted during biomass burning and hence is influenced by anthropogenic pollution (as can be seen in the second figure below, it follows the trends of the pollution), and hence, it also will not serve as a tracer for mineral dust. It should also be mentioned, that we only examined the soluble fractions of these components, while in dust particles, they might occur in non-soluble compounds.

Furthermore, INP make up a very small amount of both, total mass and total number concentration (a general value that is often given is that one in a million particles is ice active at -20°C), so it might be impossible to correlate atmospheric INP with a chemical compound based on the chemical analysis we did here.

And last but not least, the broad temperature range over which INP concentrations in the atmosphere are observed to increase generally suggest that a number of different types of INP participate (it is often assumed that more rarely occurring biological particles cause ice nucleation at higher temperatures ( >-15°C) while mineral dust particles are responsible for the observed ice nucleation at lower temperatures), so we might have to look at a combination of different trace components.

But all this was not the aim of this study. We wanted to see if anthropogenic pollution might add INP, which we showed is not the case. Still, we added a paragraph summarizing what we answered to you above, which can be found at the end of section 3.3, from line 371 onward:

*"Additionally, also no correlation was found between any of the water-soluble constituents that were analyzed with ion chromatography and INP concentrations. This is not too astounding, as INP make up only a small fraction of all particles, as can be seen when comparing number concentrations from Fig. 4 and Fig. 7, and hence they make up only a small fraction of the mass, likely too small to be detected. Furthermore, a number of different components might contribute to INP, e.g., biological INP that are generally ice active at higher temperatures (> -15°C, Murray et al., 2012) and mineral dusts which are ice active at lower temperatures, therefore one common tracer for INP might not be applicable. As far as K is concerned, which might be connected to K-feldspar containing mineral dust particles with high ice activity (Atkinson et al., 2013), we only analyzed the water soluble fraction, i.e., K related to feldspar would not have been analyzed. Moreover, K is also emitted by biomass burning and hence influenced by anthropogenic pollution. It remains to be seen if a simple correlation between chemical constituents of the atmospheric aerosol and INP concentrations can be established at all."*

[Figure]

Figure above: Measured INP number concentrations at -16°C plotted versus $Ca^{2+}$ mass concentrations derived from ion chromatography analyzing water soluble ions.

[Figure]

Figure above: The time series of measured $N_{INP}$ and $N_{INP}$ parameterized according to DeMott et al. (2010, 2015) at -16ºC, together with $Ca^{2+}$ and $K^+$ mass concentrations derived from ion chromatography analyzing water soluble ions.

Literature:

Atkinson, J. D., B. J. Murray, M. T. Woodhouse, T. F. Whale, K. J. Baustian, K. S. Carslaw, S. Dobbie, D. O'Sullivan, and T. L. Malkin (2013), The importance of feldspar for ice nucleation by mineral dust in mixed-phase clouds, Nature, 498(7454), 355-358, doi:10.1038/nature12278.

Conen, F., Henne, S., Morris, C. E., and Alewell, C.: Atmospheric ice nucleators active>=12°C can be quantified on PM10 filters, Atmos. Meas. Tech., 5, 321-327, doi:10.5194/amt-5-321-2012, 2012.

Murray, B. J., O'Sullivan, D., Atkinson, J. D., and Webb, M. E.: Ice nucleation by particles immersed in supercooled cloud droplets, Chem. Soc. Rev., 41, 6519-6554, doi:10.1039/c2cs35200a, 2012.

---

## Referee Report (RR1)

**Review of "Ice nucleating particle concentrations unaffected by urban air pollution in Beijing, China" by Chen et al.**

**General Comment:**

The original manuscript has been significantly improved. Most of my previous concerns were nicely addressed in the revised version. The reviewer has three additional comments that would like to clarify before the manuscript is accepted for its publications in ACP.

**Additional comments:**

1. The reviewer is surprised the authors completely ignored meteorology in this study. A detailed analysis of the meteorological variables and air masses is required to explain ambient observations, even in urban areas.

Authors: We added two plots showing trajectories and also wind direction and wind speed, to show the meteorological condition during the sampling period, together with the following text (line 234-245):

*Additionally, Fig.2 shows 2-day back-trajectories obtained by the NOAA HYSPLIT model, with one trajectory related to each sampled filter, starting at the median sampling time of each filter. Fig. 3 shows minutely recorded data for wind-direction and wind–speed collected by (Met One 591) and (Met One 590) located on the same roof top as the aerosol sampling equipment. Both pictures are colored-coded with respect to PM2.5 mass concentrations. The air masses that came from north or north-western directions were generally coincident with higher wind-speeds. They brought clean air with lower PM2.5 mass concentrations. They did cross desert regions, however, Beijing was reported to be affected by desert dust in mainly only spring (Wu et al., 2009). Typically, the air masses coming from south and south-west of Beijing moved slowly and spent much more time over industrialized regions, resulting in high particulate matter mass concentrations. This here observed pattern is typical for Beijing, and these connections between wind-direction and pollution levels in Beijing have been analyzed in detail previously in Wehner et al. (2008)."*

Reviewer: The reviewer appreciate the addition of the new plots and new text. This is very useful information for the readers. However, what is still missing is the correlation of the

meteorological variables with the INP concentrations. I assume that one of the main goals of this study is to identify the source of the INPs. Meteorology could help the authors to understand this. Is the INPs concentration lower or higher when the air masses were from the north and north-west with low PM2.5? Is the INPs concentration lower or higher when the air masses were from the south and south-west with high PM2.5? How about wind speed? Does it have any influence in the INP concentration?

2. The authors claim that the measured INPs are non-urban and they suggests that the sources of the INPs could be dust or bioparticles which are non-urban. Do the authors think that is it not possible to have urban dust and urban bioparticles?

Authors: Indeed, there may be urban dust and also urban bioparticles. But these are no major contributor to the increase in PM2.5 mass concentrations during winter times – rather, it is well known that this increase is related to anthropogenic pollution. On the other hand, considering biogenic and dust particles, these particles emitted from urban areas will only contribute little to the overall atmospheric dust and biogenic particle load, as the non-urban sources are much more dominant for these types of particles. Therefore explicitly mentioning that dust and biogenic particles might also be emitted from urban sources does not really make sense. If there is, however, a specific passage in the text that you feel is miss-formulated, please tell us where this is exactly and why precisely you think this is wrong. For the time being, nothing was changed.

Reviewer: Here I disagree with the response. The authors agree that there are urban dust and bioparticles; however, they provide two arguments to say these particles are not important. It is said that i) they "are no major contributor to the increase in PM2.5 mass concentrations during winter times" and ii) that "particles emitted from urban areas will only contribute little to the overall atmospheric dust and biogenic particle load". First, I agree that they are not a major contributor to PM2.5, but this is not a good argument because PM2.5 was not found to correlate with the INP concentrations. Second, although the concentration of urban bioparticles could be small, if they have good ice nucleating abilities, they can significantly contribute to the INP concentrations. I don't clearly understand why urban bioparticles are negligible and do not make sense to the authors.

3. It is unclear how BC was calculated/determined for the PM2.5 reported in Figure 1.

Authors: The BC was measured by a multi-angle absorption photometer (5012, MAAP, Thermo Fisher Scientific, Waltham, MA, USA) which got the sampled air through an inlet with a 2.5μm cut-off. This had been included in the previous version of the manuscript. The instrument measures the absorption of particles collected on a filter with a time resolution of 5 min and automatically derives BC mass concentration from the measurement while accounting for multiple scattering occurring on the filter. The MAAP is a well known and often used instrument for the measurement of absorption coefficients and BC mass concentrations.

The respective retrieval of BC values is now added in line 143 ff:"*The instrument measures the absorption of particles collected on a filter with a time resolution of 5 min and automatically derives BC mass concentration from the measurement while accounting for multiple scattering occurring on the filter.*"

Reviewer: The reviewer is very familiar with the MAAP and perhaps my questions was not well formulated. If I understood correctly the concentration of the ions was obtained from the PTEF filters, right? But what it is unclear to me is if the BC concentration reported in Figure 1 is obtained directly from the MAAP or if this was obtained off-line (using other technique), similar to the ions concentrations. If the reported BC data in Figure 1 is from the MAAP, is it directly comparable to ions off-line data?

---

## Author Response (AR2)

**Answers to the report by anonymous Referee #1**

We thank Referee 1 for reviewing our manuscript and also for useful hints and suggestions. Below, comments from the referee are given in blue while our new answers are given in purple, with passages including new text given in italic. Additionally, the new text is marked yellow in the revised version of the manuscript.

General Comment:
The original manuscript has been significantly improved. Most of my previous concerns were nicely addressed in the revised version. The reviewer has three additional comments that would like to clarify before the manuscript is accepted for its publications in ACP.

Major Comments:
1. The reviewer is surprised the authors completely ignored meteorology in this study. A detailed analysis of the meteorological variables and air masses is required to explain ambient observations, even in urban areas.

Authors: We added two plots showing trajectories and also wind direction and wind speed, to show the meteorological condition during the sampling period, together with the following text (line 234-245):
Additionally, Fig.2 shows 2-day back-trajectories obtained by the NOAA HYSPLIT model, with one trajectory related to each sampled filter, starting at the median sampling time of each filter. Fig. 3 shows minutely recorded data for wind-direction and wind–speed collected by (Met One 591) and (Met One 590) located on the same roof top as the aerosol sampling equipment. Both pictures are colored-coded with respect to PM2.5 mass concentrations. The air masses that came from north or north-western directions were generally coincident with higher wind-speeds. They brought clean air with lower PM2.5 mass concentrations. They did cross desert regions, however, Beijing was reported to be affected by desert dust in mainly only spring (Wu et al., 2009). Typically, the air masses coming from south and south-west of Beijing moved slowly and spent much more time over industrialized regions, resulting in high particulate matter mass concentrations. This here observed pattern is typical for Beijing, and these connections between wind-direction and pollution levels in Beijing have been analyzed in detail previously in Wehner et al. (2008)."

Reviewer: The reviewer appreciate the addition of the new plots and new text. This is very useful information for the readers. However, what is still missing is the correlation of themeteorological variables with the INP concentrations. I assume that one of the main goals ofthis study is to identify the source of the INPs. Meteorology could help the authors to understand this. Is the INPs concentration lower or higher when the air masses were from the north and north-west with low PM2.5? Is the INPs concentration lower or higher when the air masses were from the south and south-west with high PM2.5? How about wind speed? Does it have any influence in the INP concentration?

Authors: Our main goal was to see if anthropogenically generated particles in the polluted Beijing winter air would contribute to atmospheric INP. Still, following the reviewer's suggestions, two plots are displayed here. The first one shows the relationship between INP concentration and wind-speed. The second one displays the INP concentration-colored backward trajectories. No clear correlation between wind speed and INP concentrations was found (Fig. 1). Fig. 2 displays the backward trajectories colored by INP concentrations. There were no evident dependency of the direction of air masses and INP concentration.

We added R2 and p values for the correlation with wind speed in Table 1, added Fig 2 to the manuscript (Fig.9), and added the following text (line 368):

*"In addition to what we discussed above, also no correlation was observed between $N_{INP}$ and wind-speed, as can be seen by the respective values for $R^2$ and p given in Table 1. Fig. 9 indicates that there was also no correlation with wind-direction. The fact that we find no correlation with either wind-speed or wind-direction agrees with the desert regions towards the north-west not being efficient dust sources in winter, and are a hint that we may have observed average background INP concentrations in Beijing during our measurements."*

[Figure]

Figure 1: INP concentration vs. wind speed.

[Figure]

Figure 2: The 2-day back-trajectories obtained by the NOAA HYSPLIT model colored-coded
with respect to INP concentration

2. The authors claim that the measured INPs are non-urban and they suggests that the
sources of the INPs could be dust or bioparticles which are non-urban. Do the authors
think that is it not possible to have urban dust and urban bioparticles?

Authors: Indeed, there may be urban dust and also urban bioparticles. But these are no major
contributor to the increase in PM2.5 mass concentrations during winter times – rather, it is
well known that this increase is related to anthropogenic pollution. On the other hand,
considering biogenic and dust particles, these particles emitted from urban areas will only
contribute little to the overall atmospheric dust and biogenic particle load, as the non-urban
sources are much more dominant for these types of particles. Therefore explicitly mentioning
that dust and biogenic particles might also be emitted from urban sources does not really
make sense. If there is, however, a specific passage in the text that you feel is missformulated,
please tell us where this is exactly and why precisely you think this is wrong. For
the time being, nothing was changed.

Reviewer: Here I disagree with the response. The authors agree that there are urban dust and
bioparticles; however, they provide two arguments to say these particles are not important. It
is said that i) they "are no major contributor to the increase in PM2.5 mass concentrations
during winter times" and ii) that "particles emitted from urban areas will only contribute little to the overall atmospheric dust and biogenic particle load". First, I agree that they are not a major contributor to PM2.5, but this is not a good argument because PM2.5 was not found to correlate with the INP concentrations. Second, although the concentration of urban bioparticles could be small, if they have good ice nucleating abilities, they can significantly contribute to the INP concentrations. I don't clearly understand why urban bioparticles are negligible and do not make sense to the authors.

Authors: We cannot determine the contribution of the urban dust and bioparticles to the total aerosol load, and yes, these particles might be efficient INP and could contribute to the observed INP, although particularly biogenic particles might not be present in large amounts due to the season. We adjusted the text in line 425-433 (given in regular script here), with the following (given in italic):
"Despite the difference among methods and ice nucleating modes, this again suggests that urban *pollution* aerosol particles might not be efficient immersion freezing INP and that the ice nucleating ability of particles in urban aerosols might originate from the non-urban background aerosol particles that are included in the urban aerosol, i.e., that INP observed in urban environments might have the same sources among bioaerosols and dust particles as non-urban INP. *An additional contribution from urban biogenic or dust particles to the INP observed in this study cannot be fully excluded, but the agreement between our data and rural data presented in literature (see Fig. 8 and Table 2) corroborates our assumption that atmospheric INP in general originate from non-urban sources.*"

3. It is unclear how BC was calculated/determined for the PM2.5 reported in Figure 1.

Authors: The BC was measured by a multi-angle absorption photometer (5012, MAAP, Thermo Fisher Scientific, Waltham, MA, USA) which got the sampled air through an inlet with a 2.5μm cut-off. This had been included in the previous version of the manuscript. The instrument measures the absorption of particles collected on a filter with a time resolution of 5 min and automatically derives BC mass concentration from the measurement while accounting for multiple scattering occurring on the filter. The MAAP is a well known and often used instrument for the measurement of absorption coefficients and BC mass concentrations.
The respective retrieval of BC values is now added in line 143 ff:"The instrument measures the absorption of particles collected on a filter with a time resolution of 5 min and automatically derives BC mass concentration from the measurement while accounting for multiple scattering occurring on the filter."

Reviewer: The reviewer is very familiar with the MAAP and perhaps my questions was not well formulated. If I understood correctly the concentration of the ions was obtained from the PTEF filters, right? But what it is unclear to me is if the BC concentration reported in Figure 1 is obtained directly from the MAAP or if this was obtained off-line (using other technique), similar to the ions concentrations. If the reported BC data in Figure 1 is from the MAAP, is it directly comparable to ions off-line data?

Authors:

Typically, elemental carbon (EC) concentrations are taken to combine with filter-based ions concentrations. Here, we compared the EC concentration measured by filter-based Sunset EC/OC analyzer with BC concentration measured by MAAP during one recent field campaign. The plot is given below. The comparison showed that the BC measured by MAAP is slightly higher than EC derived from filter-based Sunset EC/OC. The mean ratio of BC/EC is about 1.35. We also added this in the manuscript, see line 147-150:

[revised manuscript text omitted]